# SyncDreamer: Generating Multiview-consistent Images from a Single-view Image

**Yuan Liu**[1,2]∗  **Cheng Lin**[2]∗  **Zijiao Zeng**[2]  **Xiaoxiao Long**[1]†  **Lingjie Liu**[3]
**Taku Komura**[1]  **Wenping Wang**[4]†
[1]The university of Hong Kong   [2]Tencent Games   [3]University of Pennsylvania
[4]Texas A&M University
{yuanly, chlin, xxlong}@connect.hku.hk   zijiao@tencent.com
lingjie.liu@seas.upenn.edu   wenping@tamu.edu

## Abstract

In this paper, we present a novel diffusion model called *SyncDreamer* that generates multiview-consistent images from a single-view image. Using pretrained large-scale 2D diffusion models, recent work Zero123 (Liu et al., 2023b) demonstrates the ability to generate plausible novel views from a single-view image of an object. However, maintaining consistency in geometry and colors for the generated images remains a challenge. To address this issue, we propose a synchronized multiview diffusion model that models the joint probability distribution of multiview images, enabling the generation of multiview-consistent images in a single reverse process. SyncDreamer synchronizes the intermediate states of all the generated images at every step of the reverse process through a 3D-aware feature attention mechanism that correlates the corresponding features across different views. Experiments show that SyncDreamer generates images with high consistency across different views, thus making it well-suited for various 3D generation tasks such as novel-view-synthesis, text-to-3D, and image-to-3D. Project page: https://liuyuan-pal.github.io/SyncDreamer/.

## 1 Introduction

Humans possess a remarkable ability to perceive 3D structures from a single image. When presented with an image of an object, humans can easily imagine the other views of the object. Despite great progress (Yao et al., 2018; Tewari et al., 2020; Wang et al., 2021; Mildenhall et al., 2020; Xie et al., 2022) brought by neural networks in computer vision or graphics fields for extracting 3D information from images, generating multiview-consistent images from a single-view image of an object is still a challenging problem due to the limited 3D information available in an image.

Recently, diffusion models (Rombach et al., 2022; Ho et al., 2020) have demonstrated huge success in 2D image generation, which unlocks new potential for 3D generation tasks. However, directly training a generalizable 3D diffusion model (Wang et al., 2023b; Jun & Nichol, 2023; Nichol et al., 2022; Müller et al., 2023) usually requires a large amount of 3D data while existing 3D datasets are insufficient for capture the complexity of arbitrary 3D shapes. Therefore, recent methods (Poole et al., 2023; Wang et al., 2023a;d; Lin et al., 2023; Chen et al., 2023b) resort to distilling pretrained text-to-image diffusion models for creating 3D models from texts, which shows impressive results on this text-to-3D task. Some works (Tang et al., 2023a; Melas-Kyriazi et al., 2023; Xu et al., 2022; Raj et al., 2023) extend such a distillation process to train a neural radiance field (Mildenhall et al., 2020) (NeRF) for the image-to-3D task. In order to utilize pretrained text-to-image models, these methods have to perform textual inversion (Gal et al., 2022) to find a suitable text description of the input image. However, the distillation process along with the textual inversion usually takes a long time to generate a single shape and requires tedious parameter tuning for satisfactory quality. Moreover, due to the abundance of specific details in an image, such as object category, appearance,

---

∗Equal contribution.
†Corresponding Authors.

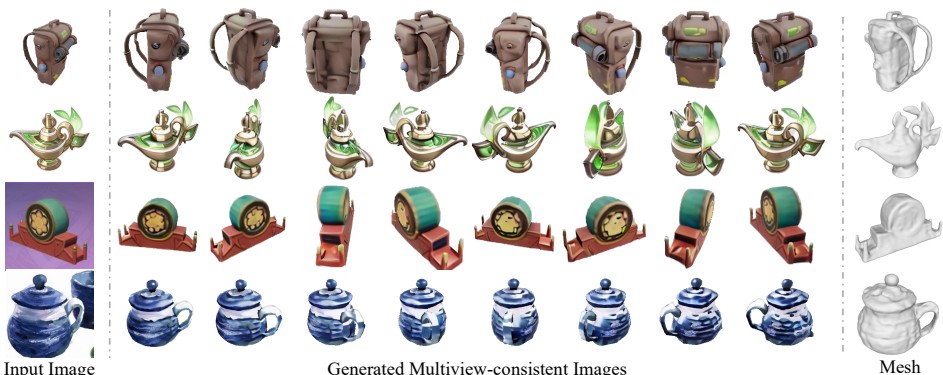

Input Image        Generated Multiview-consistent Images        Mesh

Figure 1: **SyncDreamer** is able to generate multiview-consistent images from a single-view input image of arbitrary objects. The generated multiview images can be used for mesh reconstruction by reconstruction methods like NeuS (Wang et al., 2021) without using SDS (Poole et al., 2023) loss.

and pose, it is challenging to accurately represent an image using a single word embedding, which results in a decrease in the quality of 3D shapes reconstructed by the distillation method.

Instead of distillation, some recent works (Watson et al., 2022; Gu et al., 2023b; Deng et al., 2023a; Zhou & Tulsiani, 2023; Tseng et al., 2023; Yu et al., 2023b; Chan et al., 2023; Tewari et al., 2023; Zhang et al., 2023b; Xiang et al., 2023) apply 2D diffusion models to directly generate multiview images for the 3D reconstruction task. The key problem is how to maintain the multiview consistency when generating images of the same object. To improve the multiview consistency, these methods allow the diffusion model to condition on the input images (Zhou & Tulsiani, 2023; Tseng et al., 2023; Watson et al., 2022; Liu et al., 2023b; Yu et al., 2023b), previously generated images (Tewari et al., 2023; Chan et al., 2023) or renderings from a neural field (Gu et al., 2023b). Although some impressive results are achieved for specific object categories from ShapeNet (Chang et al., 2015) or Co3D (Reizenstein et al., 2021), how to design a diffusion model to generate multiview-consistent images for arbitrary objects still remains unsolved.

In this paper, we propose a simple yet effective framework to generate multiview-consistent images for the single-view 3D reconstruction of arbitrary objects. The key idea is to extend the diffusion framework (Ho et al., 2020) to model the joint probability distribution of multiview images. We show that modeling the joint distribution can be achieved by introducing a synchronized multiview diffusion model. Specifically, for $N$ target views to be generated, we construct $N$ shared noise predictors respectively. The reverse diffusion process simultaneously generates $N$ images by $N$ corresponding noise predictors, where information across different images is shared among noise predictors by attention layers on every denoising step. Thus, we name our framework *SyncDreamer* which synchronizes intermediate states of all noise predictors on every step in the reverse process.

SyncDreamer has the following characteristics that make it a competitive tool for lifting 2D single-view images to 3D. First, SyncDreamer retains strong generalization ability by initializing its weights from the pretrained Zero123 (Liu et al., 2023b) model which is finetuned from the Stable Diffusion model (Rombach et al., 2022) on the Objaverse (Deitke et al., 2023b) dataset. Thus, SyncDreamer is able to reconstruct shapes from both photorealistic images and hand drawings as shown in Fig. 1. Second, SyncDreamer makes the single-view reconstruction easier than the distillation methods. Because the generated images are consistent in both geometry and appearance, we can simply run a vanilla NeRF (Mildenhall et al., 2020) or a vanilla NeuS (Wang et al., 2021) without using any special losses for reconstruction. Given the generated images, one can easily reckon the final reconstruction quality while it is hard for distillation methods to know the output reconstruction quality beforehand. Third, SyncDreamer maintains creativity and diversity when inferring 3D information, which enables generating multiple reasonable objects from a given image as shown in Fig. 4. In comparison, previous distillation methods can only converge to one single shape.

We quantitatively compare SyncDreamer with baseline methods on the Google Scanned Object (Downs et al., 2022) dataset. The results show that, in comparison with baseline methods, SyncDreamer is able to generate more consistent images and reconstruct better shapes from input single-view images. We further demonstrate that SyncDreamer supports various styles of 2D input

like cartoons, sketches, ink paintings, and oil paintings for generating consistent views and reconstructing 3D shapes, which verifies the effectiveness of SyncDreamer in lifting 2D images to 3D.

# 2 RELATED WORK

## 2.1 DIFFUSION MODELS

Diffusion models (Ho et al., 2020; Rombach et al., 2022; Croitoru et al., 2023) have shown impressive results on 2D image generation. Concurrent work MVDiffusion (Tang et al., 2023b) also adopts the multiview diffusion formulation to synthesize textures or panoramas with known geometry. We propose similar formulations in SyncDreamer but with unknown geometry. MultiDiffusion (Bar-Tal et al., 2023) and SyncDiffusion (Lee et al., 2023) correlate multiple diffusion models for different regions of a 2D image. Many recent works (Nichol et al., 2022; Jun & Nichol, 2023; Müller et al., 2023; Zhang et al., 2023a; Liu et al., 2023d; Wang et al., 2023b; Gupta et al., 2023; Cheng et al., 2023; Karnewar et al., 2023b; Anciukevičius et al., 2023; Zeng et al., 2022; Erkoç et al., 2023; Chen et al., 2023a; Kim et al., 2023; Ntavelis et al., 2023; Gu et al., 2023a; Karnewar et al., 2023a) try to repeat the success of diffusion models on the 3D generation task. However, the scarcity of 3D data makes it difficult to directly train diffusion models on 3D and the resulting generation quality is still much worse and less generalizable than the counterpart image generation models, though some works (Anciukevičius et al., 2023; Chen et al., 2023a; Karnewar et al., 2023b) are trying to only use 2D images for training 3D diffusion models.

## 2.2 USING 2D DIFFUSION MODELS FOR 3D

Instead of directly learning a 3D diffusion model, many works resort to using high-quality 2D diffusion models (Rombach et al., 2022; Saharia et al., 2022) for 3D tasks. Pioneer works DreamFusion (Poole et al., 2023) and SJC (Wang et al., 2023a) propose to distill a 2D text-to-image generation model to generate 3D shapes from texts. Follow-up works (Chen et al., 2023b; Wang et al., 2023d; Seo et al., 2023a; Yu et al., 2023a; Lin et al., 2023; Seo et al., 2023b; Tsalicoglou et al., 2023; Zhu & Zhuang, 2023; Huang et al., 2023; Armandpour et al., 2023; Wu et al., 2023; Chen et al., 2023c) improve such text-to-3D distillation methods in various aspects. Many works (Tang et al., 2023a; Melas-Kyriazi et al., 2023; Qian et al., 2023; Xu et al., 2022; Raj et al., 2023; Shen et al., 2023) also apply such a distillation pipeline in the single-view reconstruction task. Though some impressive results are achieved, these methods usually require a long time for textual inversion (Liu et al., 2023a) and NeRF optimization and they do not guarantee to get satisfactory results.

Other works (Watson et al., 2022; Gu et al., 2023b; Deng et al., 2023a; Zhou & Tulsiani, 2023; Tseng et al., 2023; Chan et al., 2023; Yu et al., 2023b; Tewari et al., 2023; Yoo et al., 2023; Szymanowicz et al., 2023; Tang et al., 2023b; Xiang et al., 2023; Liu et al., 2023c; Lei et al., 2022) directly apply the 2D diffusion models to generate multiview images for 3D reconstruction. (Tseng et al., 2023; Yu et al., 2023b) are conditioned on the input image by attention layers for novel-view synthesis in indoor scenes. Our method also uses attention layers but is intended for object reconstruction. (Xiang et al., 2023; Zhang et al., 2023b) resort to estimated depth maps to warp and inpaint for novel-view image generation, which strongly relies on the performance of the external single-view depth estimator. Two concurrent works (Chan et al., 2023; Tewari et al., 2023) generate new images in an autoregressive render-and-generate manner, which demonstrates good performances on specific object categories or scenes. In comparison, SyncDreamer is targeted to reconstruct arbitrary objects and generates all images in one reverse process. The concurrent work Viewset Diffusion (Szymanowicz et al., 2023) shares a similar idea to generate a set of images. The differences between SyncDreamer and Viewset Diffusion are that SyncDreamer does not require predicting a radiance field like Viewset Diffusion but only uses attention to synchronize the states among views and SyncDreamer fixes the viewpoints of generated views for better convergence. Another concurrent work MVDream (Shi et al., 2023) also proposes multiview generation for the text-to-3D task while our work aims to reconstruct shapes from single-view images.

## 2.3 OTHER SINGLE-VIEW RECONSTRUCTION METHODS

Single-view reconstruction is a challenging ill-posed problem. Before the prosperity of generative models used in 3D reconstruction, there are many works (Tatarchenko et al., 2019; Fu et al., 2021; Kato & Harada, 2019; Li et al., 2020; Fahim et al., 2021) that reconstruct 3D shapes from single-view images by regression (Li et al., 2020) or retrieval (Tatarchenko et al., 2019), which have difficulty in generalizing to new categories. Recent NeRF-GAN methods (Niemeyer & Geiger, 2021; Chan et al., 2022; Gu et al., 2021; Schwarz et al., 2020; Gao et al., 2022; Deng et al., 2023b) learn to generate

NeRFs for specific categories like human or cat faces. These NeRF-GANs achieve impressive results on single-view image reconstruction but fail to generalize to arbitrary objects. Although some recent works also attempt to generalize NeRF-GAN to ImageNet (Skorokhodov et al., 2023; Sargent et al., 2023), training NeRF-GANs for arbitrary objects is still challenging.

## 3 METHOD

Given an input view $\mathbf{y}$ of an object, our target is to generate multiview images of the object. We assume that the object is located at the origin and is normalized inside a cube of length 1. The target images are generated on $N$ *fixed* viewpoints looking at the object with azimuths evenly ranging from $0°$ to $360°$ and elevations of $30°$. To improve the multiview consistency of generated images, we formulate this generation process as a *multiview diffusion model*. In the following, we begin with a review of diffusion models (Sohl-Dickstein et al., 2015; Ho et al., 2020).

### 3.1 DIFFUSION

Diffusion models (Sohl-Dickstein et al., 2015; Ho et al., 2020) aim to learn a probability model $p_\theta(\mathbf{x}_0) = \int p_\theta(\mathbf{x}_{0:T}) d\mathbf{x}_{1:T}$ where $\mathbf{x}_0$ is the data and $\mathbf{x}_{1:T} := \mathbf{x}_1, ..., \mathbf{x}_T$ are latent variables. The joint distribution is characterized by a Markov Chain (*reverse process*)

$$p_\theta(\mathbf{x}_{0:T}) = p(\mathbf{x}_T) \prod_{t=1}^{T} p_\theta(\mathbf{x}_{t-1}|\mathbf{x}_t), \tag{1}$$

where $p(\mathbf{x}_T) = \mathcal{N}(\mathbf{x}_T; \mathbf{0}, \mathbf{I})$ and $p_\theta(\mathbf{x}_{t-1}|\mathbf{x}_t) = \mathcal{N}(\mathbf{x}_{t-1}; \mu_\theta(\mathbf{x}_t, t), \sigma_t^2 \mathbf{I})$. $\mu_\theta(\mathbf{x}_t, t)$ is a trainable component while the variance $\sigma_t^2$ is untrained time-dependent constants (Ho et al., 2020). The target is to learn the $\mu_\theta$ for the generation. To learn $\mu_\theta$, a Markov chain called *forward process* is constructed as

$$q(\mathbf{x}_{1:T}|\mathbf{x}_0) = \prod_{t=1}^{T} q(\mathbf{x}_t|\mathbf{x}_{t-1}), \tag{2}$$

where $q(\mathbf{x}_t|\mathbf{x}_{t-1}) = \mathcal{N}(\mathbf{x}_t; \sqrt{1-\beta_t}\mathbf{x}_{t-1}, \beta_t \mathbf{I})$ and $\beta_t$ are all constants. DDPM (Ho et al., 2020) shows that by defining

$$\mu_\theta(\mathbf{x}_t, t) = \frac{1}{\sqrt{\alpha_t}} \left( \mathbf{x}_t - \frac{\beta_t}{\sqrt{1-\bar{\alpha}_t}} \epsilon_\theta(\mathbf{x}_t, t) \right), \tag{3}$$

where $\alpha_t$ and $\bar{\alpha}_t$ are constants derived from $\beta_t$ and $\epsilon_\theta$ is a *noise predictor*, we can learn $\epsilon_\theta$ by

$$\ell = \mathbb{E}_{t, \mathbf{x}_0, \epsilon} \left[ \|\epsilon - \epsilon_\theta(\sqrt{\bar{\alpha}_t}\mathbf{x}_0 + \sqrt{1-\bar{\alpha}_t}\epsilon, t)\|_2 \right], \tag{4}$$

where $\epsilon$ is a random variable sampled from $\mathcal{N}(\mathbf{0}, \mathbf{I})$.

### 3.2 MULTIVIEW DIFFUSION

Applying the vanilla DDPM model to generate novel-view images separately would lead to difficulty in maintaining multiview consistency across different views. To address this problem, we formulate the generation process as a multiview diffusion model that correlates the generation of each view. Let us denote the $N$ images that we want to generate on the predefined viewpoints as $\{\mathbf{x}_0^{(1)}, ..., \mathbf{x}_0^{(N)}\}$ where suffix 0 means the time step 0. We want to learn the *joint distribution* of all these views $p_\theta(\mathbf{x}_0^{(1:N)}|\mathbf{y}) := p_\theta(\mathbf{x}_0^{(1)}, ..., \mathbf{x}_0^{(N)}|\mathbf{y})$. In the following discussion, all the probability functions are conditioned on the input view $\mathbf{y}$ so we omit $\mathbf{y}$ for simplicity.

The forward process of the multiview diffusion model is a direct extension of the vanilla DDPM in Eq. 2, where noises are added to every view independently by

$$q(\mathbf{x}_{1:T}^{(1:N)}|\mathbf{x}_0^{(1:N)}) = \prod_{t=1}^{T} q(\mathbf{x}_t^{(1:N)}|\mathbf{x}_{t-1}^{(1:N)}) = \prod_{t=1}^{T} \prod_{n=1}^{N} q(\mathbf{x}_t^{(n)}|\mathbf{x}_{t-1}^{(n)}), \tag{5}$$

where $q(\mathbf{x}_t^{(n)}|\mathbf{x}_{t-1}^{(n)}) = \mathcal{N}(\mathbf{x}_t^{(n)}; \sqrt{1-\beta_t}\mathbf{x}_{t-1}^{(n)}, \beta_t \mathbf{I})$. Similarly, following Eq. 1, the reverse process is constructed as

$$p_\theta(\mathbf{x}_{0:T}^{(1:N)}) = p(\mathbf{x}_T^{(1:N)}) \prod_{t=1}^{T} p_\theta(\mathbf{x}_{t-1}^{(1:N)}|\mathbf{x}_t^{(1:N)}) = p(\mathbf{x}_T^{(1:N)}) \prod_{t=1}^{T} \prod_{n=1}^{N} p_\theta(\mathbf{x}_{t-1}^{(n)}|\mathbf{x}_t^{(1:N)}), \tag{6}$$

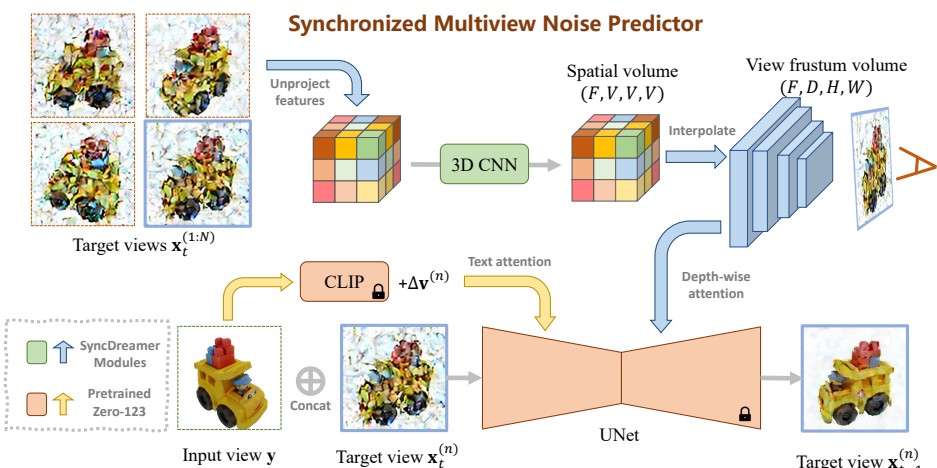

Figure 2: The pipeline of a synchronized multiview noise predictor to denoise the target view $\mathbf{x}_t^{(n)}$ for one step. First, a spatial feature volume is constructed from all the noisy target views $\mathbf{x}_t^{(1:N)}$. Then, we construct a view frustum feature volume for $\mathbf{x}_t^{(n)}$ by interpolating the features of spatial feature volume. The input view $\mathbf{y}$, current target view $\mathbf{x}_t^{(n)}$ and viewpoint difference $\Delta\mathbf{v}^{(n)}$ are fed into the backbone UNet initialized from Zero123 (Liu et al., 2023b). On the intermediate feature maps of the UNet, new depth-wise attention layers are applied to extract features from the view frustum feature volume. Finally, the output of the UNet is used to denoise $\mathbf{x}_t^{(n)}$ to obtain $\mathbf{x}_{t-1}^{(n)}$.

where $p_\theta(\mathbf{x}_{t-1}^{(n)}|\mathbf{x}_t^{(1:N)}) = \mathcal{N}(\mathbf{x}_{t-1}^{(n)}; \mu_\theta^{(n)}(\mathbf{x}_t^{(1:N)}, t), \sigma_t^2\mathbf{I})$. Note that the second equation in Eq. 6 holds because we assume a diagonal variance matrix. However, the mean $\mu_\theta^{(n)}$ of $n$-th view $\mathbf{x}_{t-1}^{(n)}$ depends on the states of all the views $\mathbf{x}_t^{(1:N)}$. Similar to Eq. 3, we define $\mu_\theta^{(n)}$ and the loss by

$$\mu_\theta^{(n)}(\mathbf{x}_t^{(1:N)}, t) = \frac{1}{\sqrt{\alpha_t}}\left(\mathbf{x}_t^{(n)} - \frac{\beta_t}{\sqrt{1-\bar{\alpha}_t}}\epsilon_\theta^{(n)}(\mathbf{x}_t^{(1:N)}, t)\right). \tag{7}$$

$$\ell = \mathbb{E}_{t,\mathbf{x}_0^{(1:N)}, n, \epsilon^{(1:N)}}\left[\|\epsilon^{(n)} - \epsilon_\theta^{(n)}(\mathbf{x}_t^{(1:N)}, t)\|_2\right], \tag{8}$$

where $\epsilon^{(1:N)}$ is the standard Gaussian noise of size $N \times H \times W$ added to all $N$ views, $\epsilon^{(n)}$ is the noise added to the $n$-th view, and $\epsilon_\theta^{(n)}$ is the noise predictor on the $n$-th view.

**Training procedure**. In one training step, we first obtain $N$ images $\mathbf{x}_0^{(1:N)}$ of the same object from the dataset. Then, we sample a timestep $t$ and the noise $\epsilon^{(1:N)}$ which is added to all the images $\mathbf{x}_0^{(1:N)}$ to obtain $\mathbf{x}_t^{(1:N)}$. After that, we randomly select a view $n$ and apply the corresponding noise predictor $\epsilon_\theta^{(n)}$ on the selected view to predict the noise. Finally, the L2 distance between the sampled noise $\epsilon^{(n)}$ and the predicted noise is computed as the loss for the training.

**Synchronized $N$-view noise predictor**. The proposed multiview diffusion model can be regarded as $N$ synchronized noise predictors $\{\epsilon_\theta^{(n)}|n=1,...,N\}$. On each time step $t$, each noise predictor $\epsilon^{(n)}$ is in charge of predicting noise on its corresponding view $\mathbf{x}_t^{(n)}$ to get $\mathbf{x}_{t-1}^{(n)}$. Meanwhile, these noise predictors are synchronized because, on every denoising step, every noise predictor exchanges information with each other by correlating the states $\mathbf{x}_t^{(1:N)}$ of all the other views. In practical implementation, we use a shared UNet for all $N$ noise predictors and put the viewpoint difference between the input view and the $n$-th target view $\Delta\mathbf{v}^{(n)}$, and the states $\mathbf{x}_t^{(1:N)}$ of all views as conditions to this shared noise predictor, i.e., $\epsilon_\theta^{(n)}(\mathbf{x}_t^{(1:N)}, t) = \epsilon_\theta(\mathbf{x}_t^{(n)}; t, \Delta\mathbf{v}^{(n)}, \mathbf{x}_t^{(1:N)})$. The detailed computation of the viewpoint difference can be found in the supplementary material.

### 3.3 3D-AWARE FEATURE ATTENTION FOR DENOISING

In this section, we discuss how to implement the synchronized noise predictor $\epsilon_\theta(\mathbf{x}_t^{(n)}; t, \Delta\mathbf{v}^{(n)}, \mathbf{x}_t^{(1:N)}, \mathbf{y})$ by correlating the multiview features using a 3D-aware attention scheme. The overview is shown in Fig. 2.

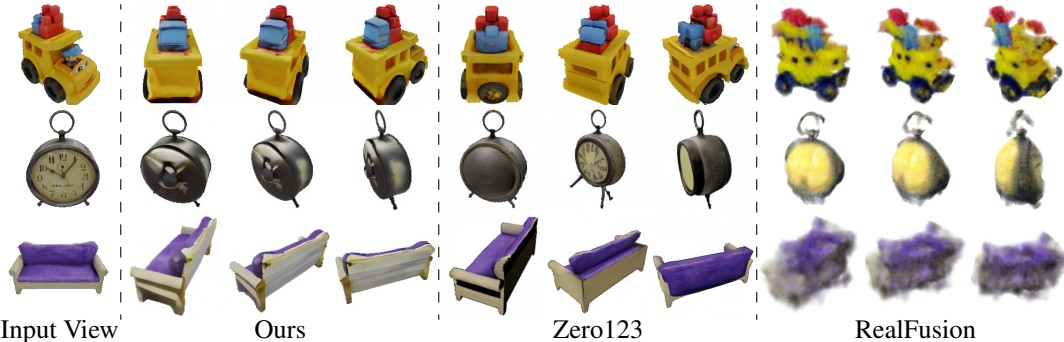

Input View          Ours          Zero123          RealFusion

Figure 3: Qualitative comparison with Zero123 and RealFusion in multiview consistency.

**Backbone UNet**. Similar to previous works (Ho et al., 2020; Rombach et al., 2022), our noise predictor $\epsilon_\theta$ contains a UNet which takes a noisy image as input and then denoises the image. To ensure the generalization ability, we initialize the UNet from the pretrained weights of Zero123 (Liu et al., 2023b) which is a generalizable model with the ability to generate novel-view images from a given image of an object. Zero123 concatenates the input view with the noisy target view as the input to UNet. Then, to encode the viewpoint difference $\Delta\mathbf{v}^{(n)}$ in UNet, Zero123 reuses the text attention layers of Stable Diffusion to process the concatenation of $\Delta\mathbf{v}^{(n)}$ and the CLIP feature (Radford et al., 2021) of the input image. We follow the same design as Zero123 and empirically freeze the UNet and the text attention layers when training SyncDreamer. Experiments to verify these choices are presented in Sec. 4.4.

**3D-aware feature attention**. The remaining problem is how to correlate the states $\mathbf{x}_t^{(1:N)}$ of all the target views for the denoising of the current noisy target view $\mathbf{x}_t^{(n)}$. To enforce consistency among multiple generated views, it is desirable for the network to perceive the corresponding features in 3D space when generating the current image. To achieve this, we first construct a 3D volume with $V^3$ vertices and then project the vertices onto all the target views to obtain the features. The features from each target view are extracted by convolution layers and are concatenated to form a spatial feature volume. Next, a 3D CNN is applied to the feature volume to capture and process spatial relationships. In order to denoise $n$-th target view, we construct a view frustum that is pixel-wise aligned with this view, whose features are obtained by interpolating the features from the spatial volume. Finally, on every intermediate feature map of the current view in the UNet, we apply a new depth-wise attention layer to extract features from the pixel-wise aligned view-frustum feature volume along the depth dimension. The depth-wise attention is similar to the epipolar attention layers in Suhail et al. (2022); Zhou & Tulsiani (2023); Tseng et al. (2023); Yu et al. (2023b) as discussed in the supplementary material.

**Discussion**. There are two primary design considerations in this 3D-aware feature attention UNet. First, the spatial volume is constructed from all the target views and all the target views share the same spatial volume for denoising, which implies a global constraint that all target views are looking at the same object. Second, the added new attention layers only conduct attention along the depth dimension, which enforces a local epipolar line constraint that the feature for a specific location should be consistent with the corresponding features on the epipolar lines of other views.

## 4 EXPERIMENTS

### 4.1 EXPERIMENT PROTOCOL

**Evaluation dataset**. Following (Liu et al., 2023b;a), we adopt the Google Scanned Object (Downs et al., 2022) dataset as the evaluation dataset. To demonstrate the generalization ability to arbitrary objects, we randomly chose 30 objects ranging from daily objects to animals. For each object, we render an image with a size of 256×256 as the input view. We additionally evaluate some images collected from the Internet and the Wiki of Genshin Impact. More results are included in the supplementary materials.

**Baselines**. We adopt Zero123 (Liu et al., 2023b), RealFusion (Melas-Kyriazi et al., 2023), Magic123 (Qian et al., 2023), One-2-3-45 (Liu et al., 2023a), Point-E (Nichol et al., 2022) and Shap-

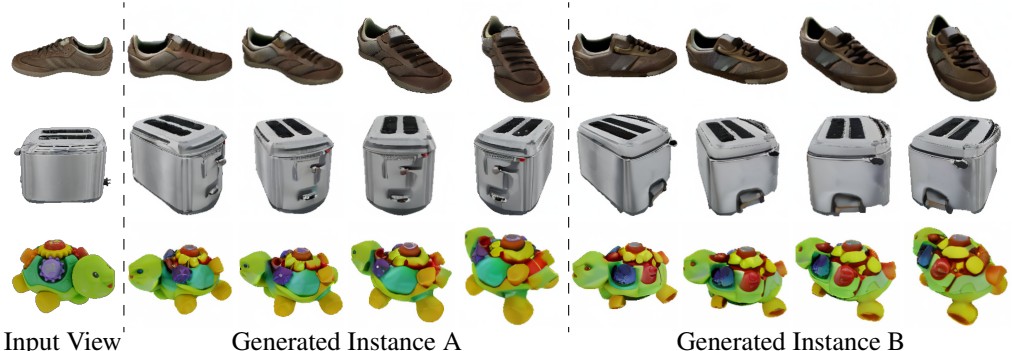

Figure 4: Different plausible instances generated by SyncDreamer from the same input image.

E (Jun & Nichol, 2023) as baseline methods. Given an input image of an object, Zero123 (Liu et al., 2023b) is able to generate novel-view images of the same object from different viewpoints. Zero123 can also be incorporated with the SDS loss (Poole et al., 2023) for 3D reconstruction. We adopt the implementation of ThreeStudio (Guo et al., 2023) for reconstruction with Zero123, which includes many optimization strategies to achieve better reconstruction quality than the original Zero123 implementation. RealFusion (Melas-Kyriazi et al., 2023) is based on Stable Diffusion (Rombach et al., 2022) and the SDS loss for single-view reconstruction. Magic123 (Qian et al., 2023) combines Zero123 (Liu et al., 2023b) with RealFusion (Melas-Kyriazi et al., 2023) to further improve the reconstruction quality. One-2-3-45 (Liu et al., 2023a) directly regresses SDFs from the output images of Zero123 and we use the official hugging face online demo (Face, 2023) to produce the results. Point-E (Nichol et al., 2022) and Shap-E (Jun & Nichol, 2023) are 3D generative models trained on a large internal OpenAI 3D dataset, both of which are able to convert a single-view image into a point cloud or a shape encoded in an MLP. For Point-E, we convert the generated point clouds to SDFs for shape reconstruction using the official models.

**Metrics**. We mainly focus on two tasks, novel view synthesis (NVS) and single view 3D reconstruction (SVR). On the NVS task, we adopt the commonly used metrics, i.e., PSNR, SSIM (Wang et al., 2004) and LPIPS (Zhang et al., 2018). To further demonstrate the multiview consistency of the generated images, we also run the MVS algorithm COLMAP (Schönberger et al., 2016) on the generated images and report the reconstructed point number. Because MVS algorithms rely on multiview consistency to find correspondences to reconstruct 3D points, more consistent images would lead to more reconstructed points. On the SVR task, we report the commonly used Chamfer Distances (CD) and Volume IoU between ground-truth shapes and reconstructed shapes. Since the shapes generated by Point-E (Nichol et al., 2022) and Shap-E (Jun & Nichol, 2023) are defined in a different canonical coordinate system, we manually align the generated shapes of these two methods to the ground-truth shapes before computing these metrics. Considering randomness in the generation, we report the min, max, and average metrics on 8 objects in the supplementary material.

## 4.2 CONSISTENT NOVEL-VIEW SYNTHESIS

For this task, the quantitative results are shown in Table 1 and the qualitative results are shown in Fig. 3. By applying a NeRF model to distill the Stable Diffusion model (Poole et al., 2023; Rombach et al., 2022), RealFusion (Melas-Kyriazi et al., 2023) shows strong multiview consistency producing more reconstructed points but is unable to produce visually plausible images as shown in Fig. 3. Zero123 (Liu et al., 2023b) produces visually plausible images but the generated images are

| Method | PSNR↑ | SSIM↑ | LPIPS↓ | #Points↑ |
|---|---|---|---|---|
| Realfusion | 15.26 | 0.722 | 0.283 | **4010** |
| Zero123 | 18.93 | 0.779 | 0.166 | 95 |
| Ours | **20.05** | **0.798** | **0.146** | 1123 |

Table 1: The quantitative comparison in novel view synthesis. We report PSNR, SSIM, LPIPS and reconstructed point numbers by COLMAP on the GSO dataset.

not multiview-consistent. Our method is able to generate images that not only are semantically consistent with the input image but also maintain multiview consistency in colors and geometry. Meanwhile, for the same input image, Our method can generate different plausible instances using different random seeds as shown in Fig. 4.

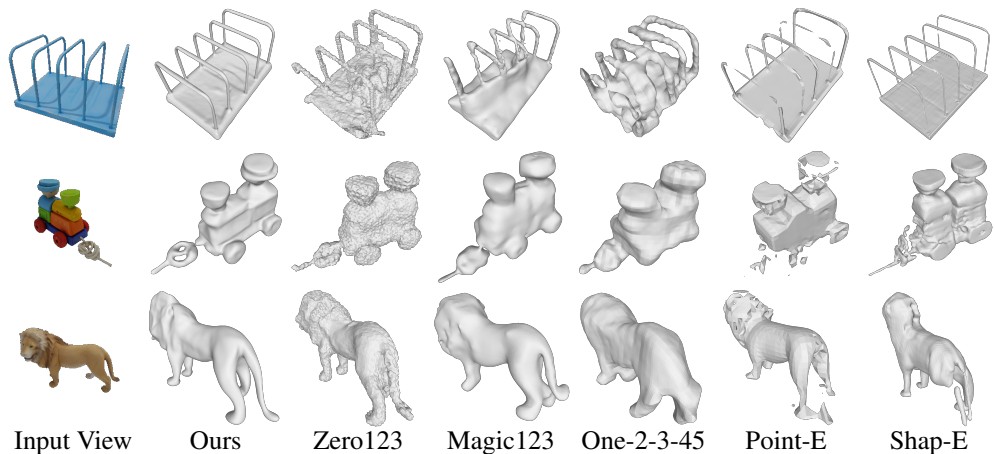

| Input View | Ours | Zero123 | Magic123 | One-2-3-45 | Point-E | Shap-E |

Figure 5: Qualitative comparison of reconstruction from single view images with different methods.

## 4.3 SINGLE VIEW RECONSTRUCTION

We show the quantitative results in Table 2 and the qualitative comparison in Fig. 5. Point-E (Nichol et al., 2022) and Shap-E (Jun & Nichol, 2023) tend to produce incompleted meshes. Directly distilling Zero123 (Liu et al., 2023b) generates shapes that are coarsely aligned with the input image, but the reconstructed surfaces are rough and not consistent with input images in detailed parts. Magic123 (Qian et al., 2023) produces much smoother meshes but heavily relies on the estimated depth values on the input view, which may lead to incorrect results when the depth estimator is not robust. One-2-3-45 (Liu et al., 2023a) reconstructs meshes from the multiview-inconsistent outputs of Zero123, which is able to capture the general geometry but also loses details. In comparison, our method achieves the best reconstruction quality with smooth surfaces and detailed geometry.

| Method | Chamfer Dist.↓ | Volume IoU↑ |
|---|---|---|
| Realfusion | 0.0819 | 0.2741 |
| Magic123 | 0.0516 | 0.4528 |
| One-2-3-45 | 0.0629 | 0.4086 |
| Point-E | 0.0426 | 0.2875 |
| Shap-E | 0.0436 | 0.3584 |
| Zero123 | 0.0339 | 0.5035 |
| Ours | **0.0261** | **0.5421** |

Table 2: Quantitative comparison with baseline methods. We report Chamfer Distance and Volume IoU on the GSO dataset.

## 4.4 DISCUSSIONS

In this section, we further conduct a set of experiments to evaluate the effectiveness of our designs.

**Generalization ability**. To show the generalization ability, we evaluate SyncDreamer with 2D designs or hand drawings like sketches, cartoons, and traditional Chinese ink paintings, which are usually created manually by artists and exhibit differences in lighting effects and color space from real-world images. The results are shown in Fig. 6. Despite the significant differences in lighting and shadow effects between these images and the real-world images, our algorithm is still able to perceive their reasonable 3D geometry and produce multiview-consistent images.

**Without 3D-aware feature attention**. To show how the proposed 3D-aware feature attention improves multiview consistency, we discard the 3D-aware attention module in SyncDreamer and train this model on the same training set. This actually corresponds to finetuning a Zero123 model with fixed viewpoints. As we can see in Fig. 7, such a model still cannot produce images with strong consistency, which demonstrates the necessity of the 3D-aware attention module in generating multiview-consistent images.

**Initializing from Stable Diffusion instead of Zero123 (Liu et al., 2023b)**. An alternative strategy is to initialize our model from Stable Diffusion (Rombach et al., 2022). However, the results shown in Fig. 7 indicate that initializing from Stable Diffusion exhibits a worse generalization ability than from Zero123. Based on our observations, we find that the batch size plays an important role in enhancing the stability and efficacy of learning 3D priors from a diverse dataset like Objaverse.

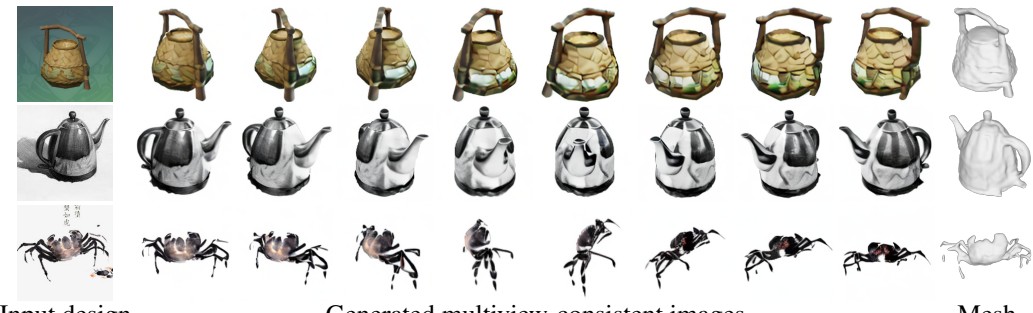

| Input design | Generated multiview-consistent images | Mesh |
|---|---|---|

Figure 6: Examples of using SyncDreamer to generate 3D models from 2D designs .

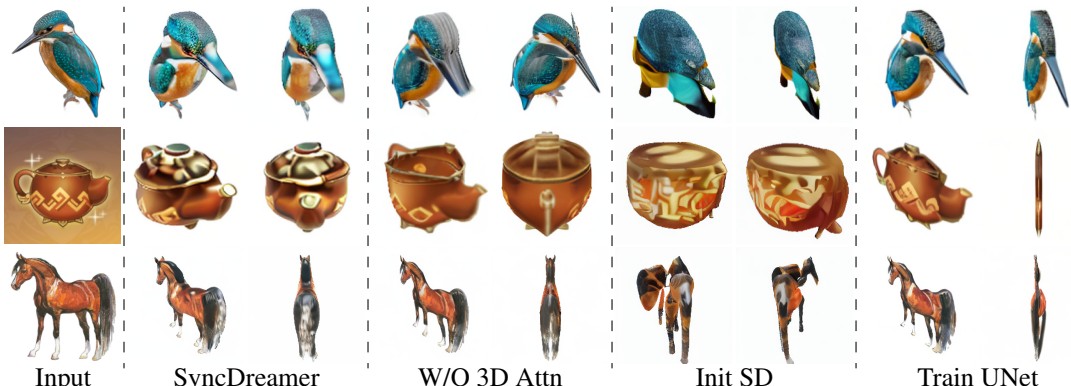

| Input | SyncDreamer | W/O 3D Attn | Init SD | Train UNet |
|---|---|---|---|---|

Figure 7: Ablation studies to verify the designs of our method. "SyncDreamer" means our full model. "W/O 3D Attn" means discarding the 3D-aware attention module in SyncDreamer, which actually results in a Zero123 (Liu et al., 2023b) finetuned on fixed viewpoints on the Objaverse (Deitke et al., 2023b) dataset. "Init SD" means initialize the SyncDreamer noise predictor from Stable Diffusion instead of Zero123. "Train UNet" means we train the UNet instead of freezing it.

However, due to limited GPU memories, our batch size is 192 which is smaller than the 1536 used by Zero123. Finetuning on Zero123 enables SyncDreamer to utilize the 3D priors of Zero123.

**Training UNet**. During the training of SyncDreamer, another feasible solution is to not freeze the UNet and the related layers initialized from Zero123 but further finetune them together with the volume condition module. As shown in Fig. 7, the model without freezing these layers tends to predict the input object as a thin plate, especially when the input images are 2D hand drawings. We speculate that this phenomenon is caused by overfitting, likely due to the numerous thin-plate objects within the Objaverse dataset and the fixed viewpoints employed during our training process.

**Runtime**. SyncDreamer uses about 40s to sample 64 images (4 instances) with 50 DDIM (Song et al., 2020) sampling steps on a 40G A100 GPU. Our runtime is slightly longer than Zero123 because we need to construct the spatial feature volume on every step.

## 5  CONCLUSION

In this paper, we present SyncDreamer to generate multiview-consistent images from a single-view image. SyncDreamer adopts a synchronized multiview diffusion to model the joint probability distribution of multiview images, which thus improves the multiview consistency. We design a novel architecture that uses the Zero123 as the backbone and a new volume condition module to model cross-view dependency. Extensive experiments demonstrate that SyncDreamer not only efficiently generates multiview images with strong consistency, but also achieves improved reconstruction quality compared to the baseline methods with excellent generalization to various input styles.

## 6 ACKNOWLEDGEMENT

This research is sponsored by the Innovation and Technology Commission of the HKSAR Government under the InnoHK initiative and Ref. T45-205/21-N of Hong Kong RGC. We sincerely thank Zhiyang Dou, Peng Wang, and Jiepeng Wang from AnySyn3D for discussions. This work is based on the computation resources from Tencent Taiji platform.

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

## A    APPENDIX

### A.1    IMPLEMENTATION DETAILS

We train SyncDreamer on the Objaverse (Deitke et al., 2023b) dataset which contains about 800k objects. We set the viewpoint number $N = 16$. The spatial volume has the size of $32^3$ and the view-frustum volume has the size of $32 \times 32 \times 48$. We sample 48 depth planes for the view-frustum volume because the view may look into the volume from the diagonal direction. We chose these sizes because the latent feature map size of an image of $256 \times 256$ in the Stable Diffusion Rombach et al. (2022) $32 \times 32$. The elevation of the target views is set to $30°$ and the azimuth evenly distributes in $[0°, 360°]$. Besides these target views, we also render 16 random views as input views on each object for training, which have the same azimuths but random elevations. We always assume that the azimuth of both the input view and the first target view is $0°$. We train the SyncDreamer for 80k steps ($\sim$4 days) with 8 40G A100 GPUs using a total batch size of 192. The learning rate is annealed from 5e-4 to 1e-5. The viewpoint difference is computed from the difference between the target view and the input view on their elevations and azimuths. Since we need an elevation of the input view to compute the viewpoint difference $\Delta\mathbf{v}^{(n)}$, we use the rendering elevation in training while we roughly estimate an elevation angle as input in inference. Note that baseline methods RealFusion (Melas-Kyriazi et al., 2023), Zero123 (Liu et al., 2023b), and Magic123 (Qian et al., 2023) all require an estimated elevation angle as input in test time. It is also possible to adopt the elevation estimator in Liu et al. (2023a) to estimate the elevation angle of the input image. To obtain surface meshes, we predict the foreground masks of the generated images using CarveKit[1]. Then, we train the vanilla NeuS (Wang et al., 2021) for 2k steps to reconstruct the shape, which costs about 10 mins. On each step, we sample 4096 rays and sample 128 points on each ray for training. Both the mask loss and the rendering loss are applied in training NeuS. The reconstruction process can be further sped up by faster reconstruction methods (Wang et al., 2023c; Guo, 2022; Wu et al., 2022) or generalizable SDF predictors (Long et al., 2022; Liu et al., 2023a) with priors.

### A.2    TEXT-TO-IMAGE-TO-3D

By incorporating text2image models like Stable Diffusion (Rombach et al., 2022) or Imagen (Saharia et al., 2022), SyncDreamer enables generating 3D models from text. Examples are shown in Fig. 8. Compared with existing text-to-3D distillation, our method gives more flexibility because users

---

[1]https://github.com/OPHoperHPO/image-background-remove-tool

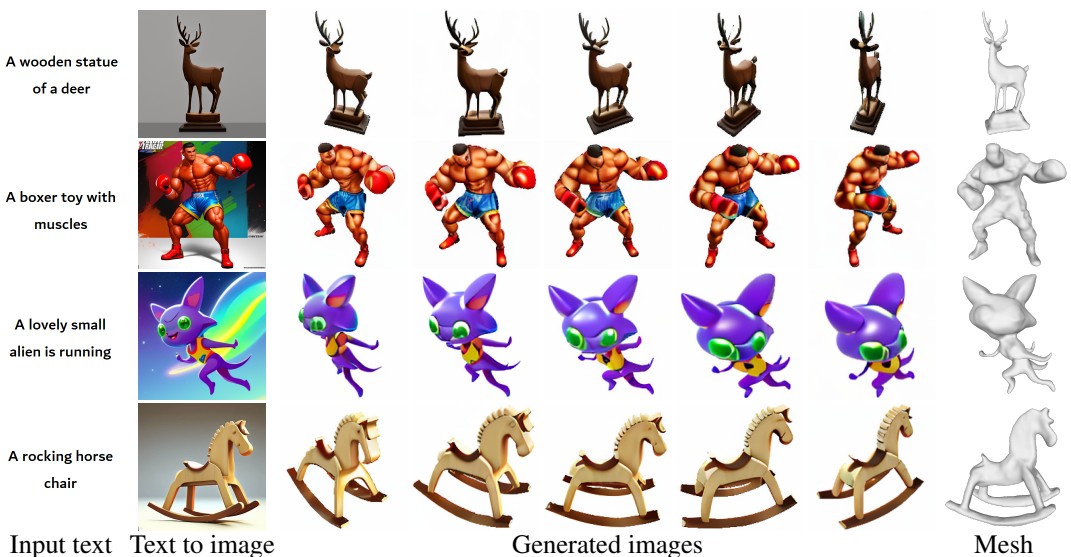

Figure 8: Examples of using SyncDreamer to generate 3D models from texts.

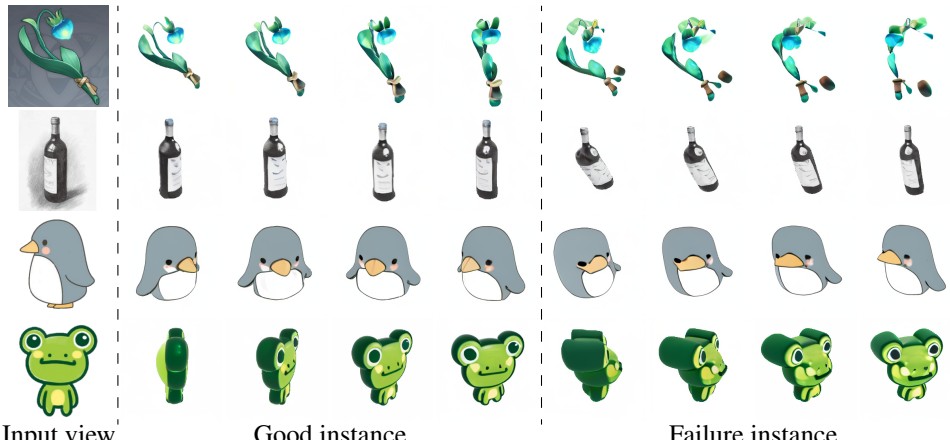

Figure 9: **Limitation on the generation quality**. The generated instances from SyncDreamer are not always desirable. Sometimes, low-quality failure instances may be generated due to the stochastic process of diffusion models.

can generate multiple images with their text2image models and select the desirable one to feed to SyncDreamer for 3D reconstruction.

## A.3 LIMITATIONS AND FUTURE WORKS

Though SyncDreamer shows promising performances in generating multiview-consistent images for 3D reconstruction, there are still limitations that the current framework does not fully address. First, the generated images of SyncDreamer have fixed viewpoints, which limits some of its application scope when requiring images of other viewpoints. A possible alternative is to use the trained NeuS to render novel-view images, which achieves reasonable but a little bit blurry results as shown in Fig. 13. Second, the generated images are not always plausible and we may need to generate multiple instances with different seeds and select a desirable instance for 3D reconstruction as shown in Fig. 9. Especially, we notice that the generation quality is sensitive to the foreground object size in the image. The reason is that changing the foreground object size corresponds to adjusting the perspective patterns of the input camera and affects how the model perceives the geometry of the object. The training images of SyncDreamer have a predefined intrinsic matrix and all are captured at

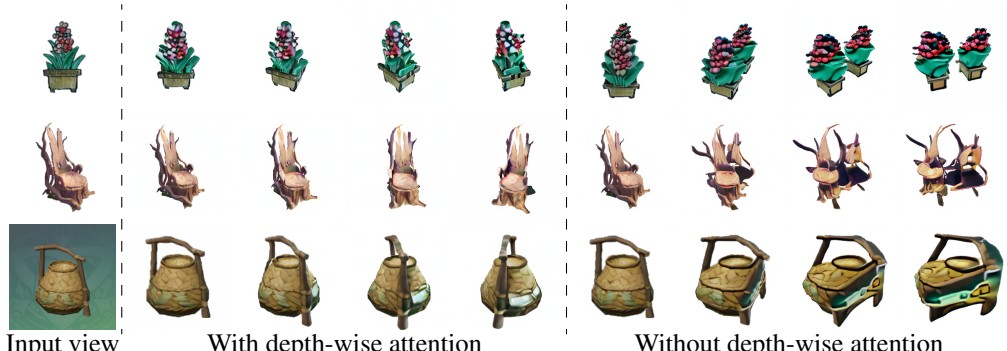

Input view      With depth-wise attention      Without depth-wise attention

Figure 10: **Evaluation of depth-wise attention layers**. The model without depth-wise attention layers has degenerated quality while the model with depth-wise attention produces better results.

a predefined distance to the constructed volume, which makes the model adapt to a fixed perspective pattern. To further increase the quality, we may need to use a larger object dataset like Objaverse-XL (Deitke et al., 2023a) and manually clean the dataset to exclude some uncommon shapes like complex scene representation, textureless 3D models, and point clouds. Third, the current implementation of SyncDreamer assumes a perspective image as input but many 2D designs are drawn with orthogonal projections, which would lead to unnatural distortion of the reconstructed geometry. Applying orthogonal projection in the volume construction of SyncDreamer would alleviate this problem. Meanwhile, we notice that generated textures are sometimes less detailed than the Zero123. The reason is that the multiview generation is more challenging, which not only needs to be consistent with the input image but also needs to be consistent with all other generated views. Thus, the model may tend to generate large texture blocks with less detail, since it could more easily maintain multiview consistency.

## A.4 DISCUSSION ON DEPTH-WISE ATTENTION LAYERS

We find that the depth-wise attention layers are important for generating high-quality multiview-consistent images. To show that, we design an alternative model that directly treats the view-frustum feature volume $H \times W \times D \times F$ as a 2D feature map $H \times W \times (D \times F)$. Then, we apply 2D convolutional layers to extract features on it and then add them to the intermediate feature maps of UNet. We find that the model without depth-wise attention layers produces degenerated images with undesirable shape distortions as shown in Fig. 10.

## A.5 GENERATING IMAGE ON OTHER VIEWPOINTS

To show the ability of SyncDreamer to generate images of different viewpoints, we train a new SyncDreamer model but with different 16 viewpoints. The new viewpoints all have elevations of $0°$ and azimuth evenly distributed in the range $[0°, 360°]$. The generated images of this new model are shown in Fig. 11.

## A.6 ITERATIVE GENERATION

It is also possible to re-generate novel view images from one of the generated images of Sync-Dreamer. Two examples are shown in Fig. 12. In the figure, row 1 shows the generated images of SyncDreamer and row 2 shows the re-generated images of SyncDreamer using one of first-row images as its input image. Though the regenerated images are still plausible, they reasonably differ from the original input view

## A.7 NOVEL-VIEW RENDERINGS OF NEUS

Though SyncDreamer can only generate images on fixed viewpoints, we can render novel-view images from arbitrary viewpoints using the NeuS model trained on the output of SyncDreamer,

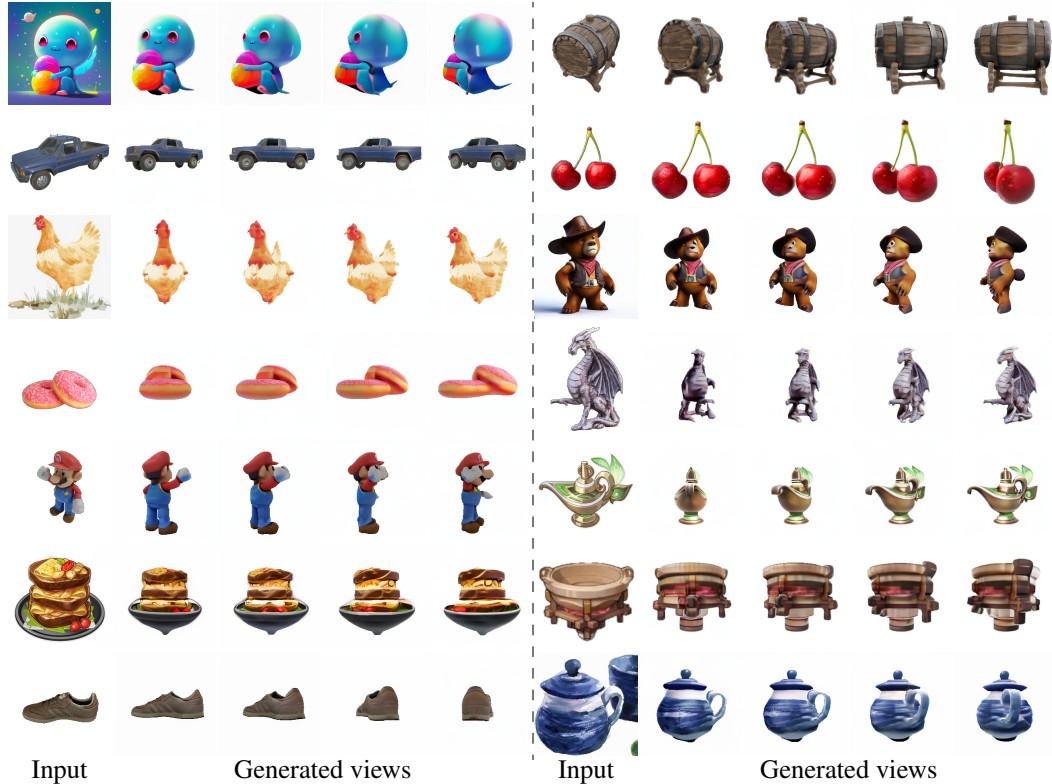

Figure 11: Generated images with 0° elevations by SyncDreamer.

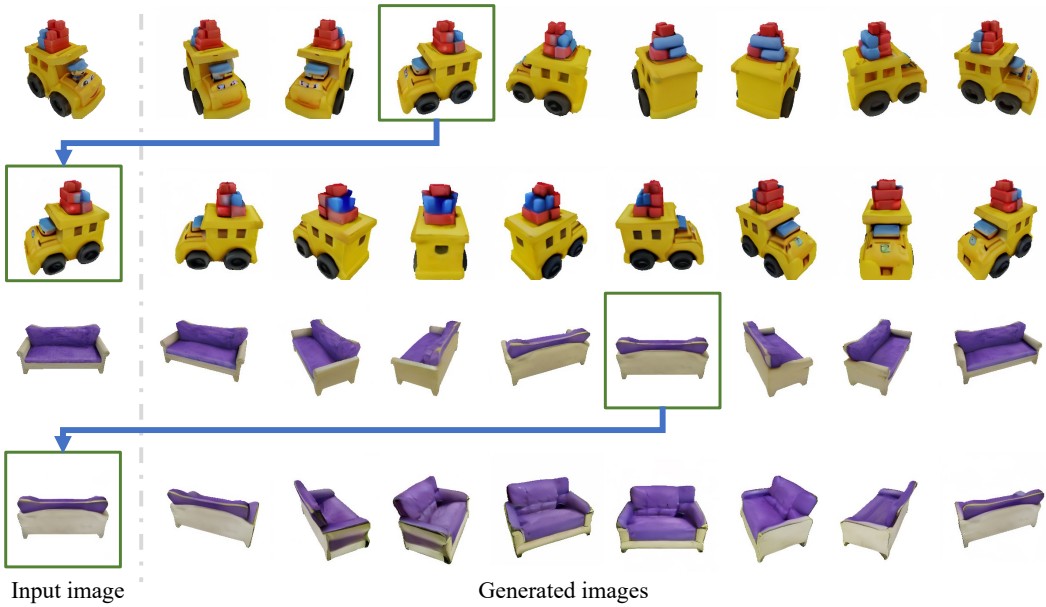

Figure 12: We use SyncDreamer to regenerate novel-view images from the outputs of SyncDreamer. For each object, Row 1 is the original generation results while Row 2 uses one output image of Row 1 as input to regenerate novel-view images.

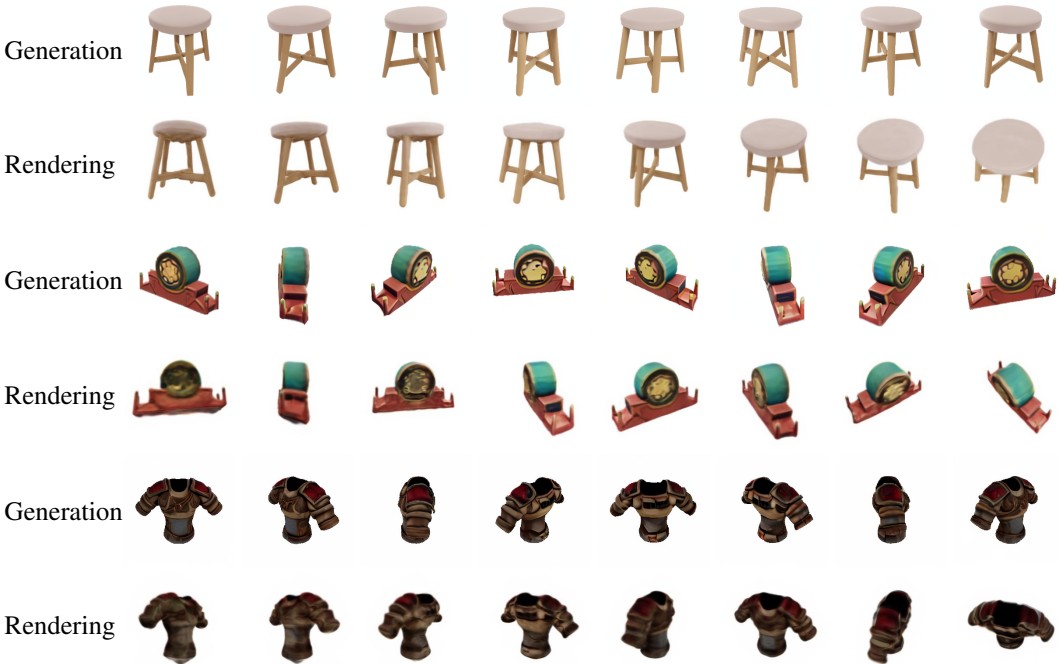

Figure 13: *Odd rows* show the image generated by SyncDreamer while *even rows* show the images rendered from arbitrary viewpoints using the NeuS model trained on the generated images.

as shown in Fig. 13. However, since only 16 images are generated to train the NeuS model, the renderings from NeuS are more blurry than the generated images of SyncDreamer.

## A.8 FEWER GENERATED VIEWS FOR NEUS TRAINING

The NeuS reconstruction process can be accomplished with fewer views, as demonstrated in Fig. 14. Decreasing the number of views from 16 to 8 does not have a significant impact on the overall reconstruction quality. However, utilizing only 4 views results in a steep decline in both surface reconstruction and novel-view-synthesis quality. Consequently, it is possible to train a more efficient version of SyncDreamer to generate 8 views for the NeuS reconstruction without compromising the quality too much.

## A.9 FASTER RECONSTRUCTION WITH HASH-GRID-BASED NEUS

It is possible to use a hash-grid-based NeuS to improve the reconstruction efficiency. Some qualitative reconstruction results are shown in Fig. 15. The hash-grid-based method takes about 3 minutes which is less than half the time of the vanilla MLP-based NeuS (10min). Since hash-grid-based SDF usually produces more noisy surfaces than MLP-based SDF, we add additional smoothness losses on the normals computed from the has-grid-based SDF.

## A.10 METRICS USING DIFFERENT GENERATION SEEDS

Due to the randomness in the generation process, the computed metrics may differ if we use different seeds for generation. To show this, we randomly sample 4 instances from the same input image of 8 objects from the GSO dataset and compute the corresponding PSNR, SSIM, LPIPS, Chamfer Distance, and Volume IOU as reported in Table 3.

4 views                8 views                16 views

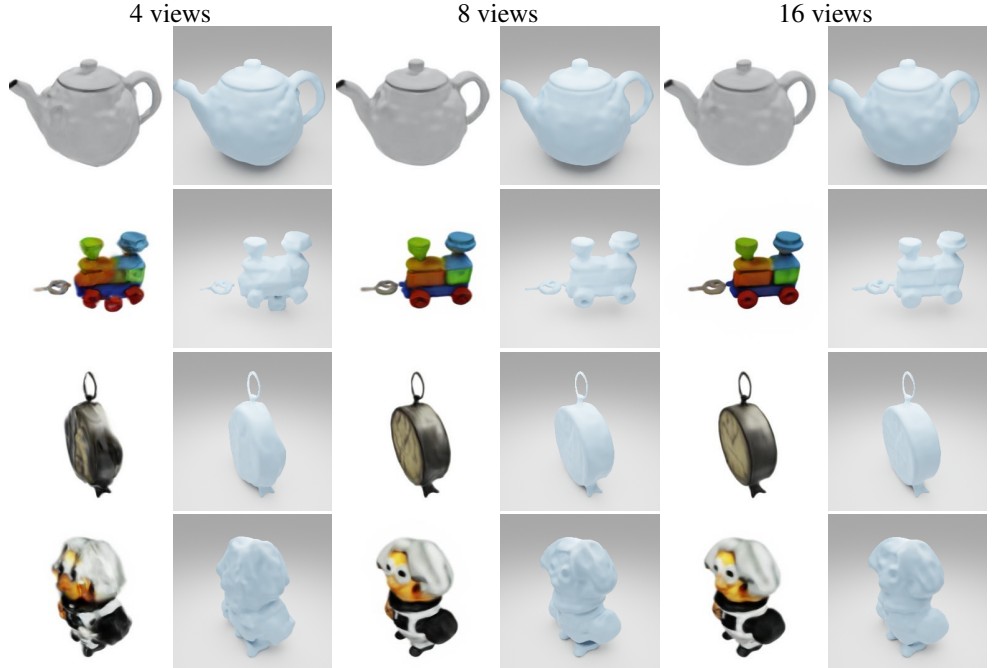

Figure 14: Results of using fewer generated views of SyncDreamer for NeuS reconstruction. *Odd columns* show the renderings of NeuS while *even columns* show the reconstructed surfaces of NeuS.

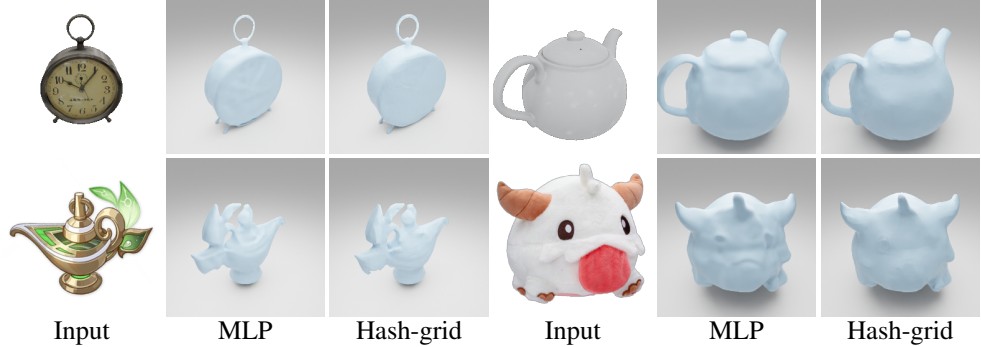

Input          MLP          Hash-grid          Input          MLP          Hash-grid

Figure 15: Surface reconstruction results using "MLP" NeuS and "Hash-grid" NeuS.

|  |  | Mario | S.Bus1 | S.Bus2 | Shoe | S.Cups | Sofa | Hat | Turtle |
|---|---|---|---|---|---|---|---|---|---|
| PSNR↑ | Min | 18.25 | 20.52 | 16.39 | 21.38 | 23.43 | 18.97 | 20.87 | 15.83 |
|  | Max | 18.74 | 20.70 | 16.67 | 21.70 | 24.48 | 19.53 | 21.08 | 16.33 |
|  | Avg. | 18.48 | 20.63 | 16.48 | 21.48 | 23.99 | 19.26 | 20.96 | 16.03 |
| SSIM↑ | Min | 0.811 | 0.851 | 0.687 | 0.862 | 0.899 | 0.809 | 0.797 | 0.749 |
|  | Max | 0.816 | 0.855 | 0.690 | 0.866 | 0.913 | 0.816 | 0.801 | 0.754 |
|  | Avg. | 0.813 | 0.853 | 0.688 | 0.864 | 0.906 | 0.812 | 0.799 | 0.751 |
| LPIPS↓ | Min | 0.129 | 0.104 | 0.222 | 0.081 | 0.055 | 0.154 | 0.134 | 0.209 |
|  | Max | 0.135 | 0.108 | 0.229 | 0.084 | 0.087 | 0.157 | 0.136 | 0.223 |
|  | Avg. | 0.133 | 0.105 | 0.226 | 0.082 | 0.071 | 0.156 | 0.135 | 0.218 |
| CD↓ | Min | 0.0139 | 0.0076 | 0.0217 | 0.0167 | 0.0079 | 0.0237 | 0.0464 | 0.0225 |
|  | Max | 0.0194 | 0.0100 | 0.0236 | 0.0184 | 0.0138 | 0.0449 | 0.0511 | 0.0377 |
|  | Avg. | 0.0167 | 0.0087 | 0.0227 | 0.0172 | 0.0110 | 0.0312 | 0.0490 | 0.0301 |
| Vol. IOU↑ | Min | 0.6604 | 0.8284 | 0.5247 | 0.4383 | 0.5966 | 0.3905 | 0.2614 | 0.6313 |
|  | Max | 0.7336 | 0.8335 | 0.5731 | 0.4826 | 0.6873 | 0.5205 | 0.2919 | 0.7471 |
|  | Avg. | 0.6889 | 0.8309 | 0.5578 | 0.4575 | 0.6427 | 0.4729 | 0.2705 | 0.6864 |

Table 3: **Statistical analysis of the generation randomness of SyncDreamer**. We generate 4 instances using SyncDreamer and compute the PSNR, SSIM, LPIPS, Chamfer Distance (CD), and Volume IOU (Vol. IOU). We list the minimum, maximum, and average values of these metrics.

## A.11 DISCUSSION ON OTHER ATTENTION MECHANISM

There are several attention mechanisms similar to our depth-wise attention layers. MVDiffusion Tang et al. (2023b) utilizes a correspondence-aware attention layer based on the known geometry. In SyncDreamer, the geometry is unknown so we cannot build such one-to-one correspondence for attention. An alternative way is the epipolar attention layer in Suhail et al. (2022); Zhou & Tulsiani (2023); Tseng et al. (2023); Yu et al. (2023b) which constructs an epipolar line on every image and applies attention along the epipolar line. Epipolar line attention constructs epipolar lines on every image and applies attention along epipolar lines. Our depth-wise attention is very similar to epipolar line attention. If we project a 3D point in the view frustum onto a neighboring view, we get a 2D sample point on the epipolar line. We notice that in epipolar line attention, we still need to maintain a new tensor of size $H \times W \times D$ containing the epipolar features. This would cost as large GPU memory as our volume-based attention. A concurrent work MVDream (Shi et al., 2023) applies attention layers on all feature maps from multiview images, which also achieves promising results. However, applying such an attention layer to all the feature maps of 16 images in our setting costs unaffordable GPU memory in training. Finding a suitable network design for multiview-consistent image generation would still be an interesting and challenging problem for future work.

## A.12 DIAGRAM ON MULTIVIEW DIFFUSION

We provide a diagram in Fig. 16 to visualize the derivation of the proposed multiview diffusion in Sec. 3.2 in the main paper.

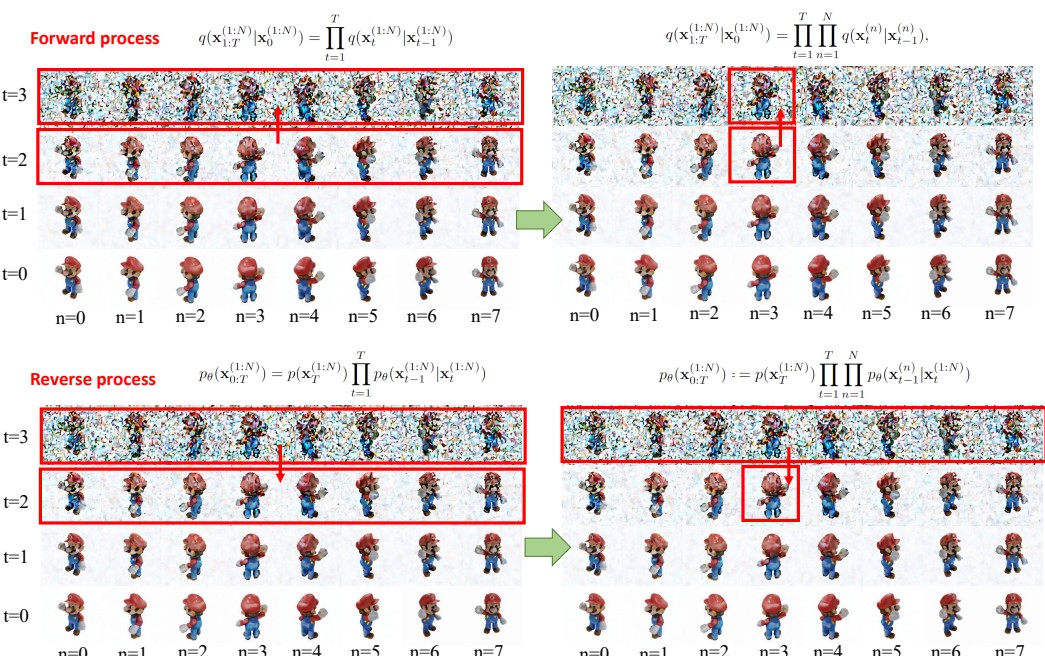

Figure 16: An intuitive diagram illustrating the derivation of the forward and reverse processes of the proposed multiview diffusion model. Better visualization quality with zooming in.

