# OpenReview forum: "SyncDreamer: Generating Multiview-consistent Images from a Single-view Image"
_ICLR.cc/2024/Conference — ICLR 2024 spotlight_

### Official Review · Reviewer_3H3E · 2023-10-27

**Soundness:** 4 excellent
**Presentation:** 3 good
**Contribution:** 4 excellent
**Rating:** 8
**Confidence:** 1

**Summary:**

The paper presents a method that is able to generate multiview-consistent images. The images can be used to reconstruct 3D model with Neus.

**Strengths:**

1. The motivation is strong and clear. The multi-view images generate by zero-1-to-3 is not consistent. The method injects 3D joint probability distribution of multiview image into zero-1-to3. The images are denoised in parallel for global awareness.
2. The experiment is thorough and convincing. The generated images achieve better PSNR and more COLMAP points than zero-1-to-3. The generated images can be used for reconstruction.
3. The presentation is clear. Good visualization and well-written sentences.

**Weaknesses:**

1. There are too many equations. It's better to use fewer equations and more intuitive sentences for method description for application papers.
2. Why the generated image is not always plausible? What's the success rate for syncdreamer?
3. No failure case visualization.

**Questions:**

1. Why using stable diffusion initialization is worse than zero-1-to-3 initialization?
2. Do you try Neus2 instead of Neus? Will Neus2 improve the reconstruction results in terms of time and accuracy?

---

> ### Author Response · Authors · 2023-11-18
>
> Thank you for your helpful comments! Our replies to your questions and revisions to the submission are stated below.
>
> **W1**: There are too many equations. It's better to use fewer equations and more intuitive sentences for method description for application papers.
>
> **A1**: Thanks for your advice. Following your suggestion, in the supplementary material, we further provide a diagram in Fig. 9 to intuitively show the derivation of the multiview diffusion process.
>
> **W2**: Why the generated image is not always plausible? What's the success rate for syncdreamer?
>
> **A2**: Generating multiview images from a single-view input is an ill-posed problem and accurately learning the joint distribution of multiview images is still a challenging task so the generated images are not plausible. Though SyncDreamer shows some promising results, its generation ability is still limited by training on the Objaverse dataset with thousands or millions of objects, which is still smaller than 2D datasets with billions of images.
>
> Exactly defining whether a generation result is successful or not is ambiguous and difficult because the generation should give a plausible distribution not a deterministic answer. Instead, we report the min/max/average metrics on the GSO dataset of our method in Table 1 of the supplementary material, which may indicate that SyncDreamer successfully generates good examples of these objects. Exactly evaluating the generation ability on the single view reconstruction task is still an unexplored and interesting problem.
>
> **W3**: No failure case visualization.
>
> **A3**: Some failure generations are shown in Figure 2 of the supplementary material.
>
> **Q1**: Why using stable diffusion initialization is worse than zero-1-to-3 initialization?
>
> **A4**: Finetuning from the Stable Diffusion model still produces multiview-consistent images but with worse quality than finetuning from Zero123. Based on our observations, we find that the batch size plays an important role in enhancing the stability and efficacy of learning 3D priors from a diverse dataset like Objaverse. However, due to limited GPU memories, our batch size is set to 192 which is much smaller than the 1536 used by Zero123. Therefore, we choose to rely on Zero123 to leverage its strong 3D priors trained with large batch sizes. If we use advanced training strategies like accumulating gradients or using more GPUs, finetuning from Stable Diffusion may produce improved results. We included an additional explanation in Sec. 4 of the main paper.
>
> **Q2**: Do you try Neus2 instead of Neus? Will Neus2 improve the reconstruction results in terms of time and accuracy?
>
> **A5**: Thanks for this valuable suggestion. NeuS2 mainly relies on the hash-grid representation to speed up the training speed. We have tried hash-grid-based NeuS in Sec. 1.9 and Fig. 8 of the supplementary material. Using a hash-grid-based NeuS indeed improves the training speed from 10 minutes to about 3 minutes with visually similar quality.

---

> > ### Comment · Reviewer_3H3E · 2023-11-22
> > **Thanks for your answer.**
> >
> > I will keep my score.

---

### Official Review · Reviewer_hhwq · 2023-10-29

**Soundness:** 4 excellent
**Presentation:** 4 excellent
**Contribution:** 4 excellent
**Rating:** 10
**Confidence:** 4

**Summary:**

This paper introduces a novel approach that suggests generating multiple views (16 views in the model) of images simultaneously using a diffusion process. The model is built upon the foundation of zero123. It puts forward a volume-based attention mechanism to facilitate communication among the various views. In terms of performance, it stands out with exceptional results, particularly excelling in multiview alignment. Remarkably, this paper marks a significant milestone as the first diffusion based approach (to the best of my knowledge) among all single-view 3D generation models that doesn't require SDS for 3D model creation.

**Strengths:**

1. The paper addresses a crucial problem in the field by tackling the issue of multiview inconsistency in generated images. This is a significant challenge as inconsistent images often require additional techniques like SDS to derive 3D information. SyncDreamer's ability to generate multiview-consistent images directly addresses this problem, reducing the need for such supplementary methods.

2. The proposed volume-based approach in SyncDreamer is a novel and innovative design. It effectively incorporates geometric regularization into the model, which enhances the generation of multiview-consistent images. The design maintains an elegant and straightforward implementation, making it a notable strength of the paper.

3. The performance of SyncDreamer is highly impressive. It stands out as the first paper in the field to enable image-to-3D conversion without relying on techniques like SDS. This achievement is attributed to the model's ability to generate images with strong multiview consistency. SyncDreamer's capacity to produce high-quality and consistent results represents a significant advancement in 3D generation tasks, making it a remarkable contribution to the field.

**Weaknesses:**

1. One notable weakness of SyncDreamer is its computational cost, particularly the significant memory requirements associated with the volume-based approach. This could pose scalability challenges when aiming for high-resolution image generation, which is an important consideration for practical applications.

2. A noticeable gap in the level of detail between images generated by Zero123 and SyncDreamer raises questions about the specifics of this discrepancy. Given that SyncDreamer leverages Zero123 as a pretrained model and maintains it frozen during training, it would be valuable to investigate whether this difference is influenced by the volume-based attention mechanism or other factors, providing a deeper understanding of the model's limitations.

3. The fact that SyncDreamer is fine-tuned on Zero123 raises concerns about its generalizability and performance when trained from scratch, especially considering that both models use the same datasets. If SyncDreamer heavily relies on a strong backbone like Zero123 to achieve its results, its broader applicability might be limited, and it would be beneficial to explore how well it performs without the pretraining.

4. The concatenation of features across all views in the volume, without a mechanism to handle occluded views, might lead to suboptimal performance, particularly in cases involving heavily occluded objects. A more refined approach to selectively attending to relevant views during convolutional operations could enhance the model's robustness.

**Questions:**

1. The paper chooses to use 16 views for its volume-based approach. It would be helpful to understand the rationale behind this specific choice. Is 16 views the minimal number required for effective multiview image reconstruction, or was this number selected for a different reason? Given the extra computational burden associated with more views, why not start with a smaller number, like 8 views, to assess the trade-off between computational efficiency and performance?

2. I tried the model and found its sensitivity to the size of objects when using the code and released models. Is this sensitivity a result of the volume-based approach or if there are specific reasons behind this behavior. Understanding the limitations related to object size could provide insights into potential improvements or workarounds. Please see below for the attached images: if I scale the 'monkey' a little bit more, the generation could be very different.

3. Epipolar-based attention mechanisms, as demonstrated in works like Generalizable Patch-Based Neural Rendering (ECCV 2022), are technically equivalent to volume-based attention. Did the authors consider or investigate this direction, and do they have any comparative results or insights regarding the choice of attention mechanism in SyncDreamer?

---
`1.jpg` and `2.jpg` show very different results when the input object (trim_size) is made slightly larger.
Please run the following python code to see the images.
```
import base64
from io import BytesIO
im = Image.open(BytesIO(base64.b64decode(b'/9j/4AAQSkZJRgABAQAAAQABAAD/2wBDAAgGBgcGBQgHBwcJCQgKDBQNDAsLDBkSEw8UHRofHh0aHBwgJC4nICIsIxwcKDcpLDAxNDQ0Hyc5PTgyPC4zNDL/2wBDAQkJCQwLDBgNDRgyIRwhMjIyMjIyMjIyMjIyMjIyMjIyMjIyMjIyMjIyMjIyMjIyMjIyMjIyMjIyMjIyMjIyMjL/wAARCAGAAIADASIAAhEBAxEB/8QAHwAAAQUBAQEBAQEAAAAAAAAAAAECAwQFBgcICQoL/8QAtRAAAgEDAwIEAwUFBAQAAAF9AQIDAAQRBRIhMUEGE1FhByJxFDKBkaEII0KxwRVS0fAkM2JyggkKFhcYGRolJicoKSo0NTY3ODk6Q0RFRkdISUpTVFVWV1hZWmNkZWZnaGlqc3R1dnd4eXqDhIWGh4iJipKTlJWWl5iZmqKjpKWmp6ipqrKztLW2t7i5usLDxMXGx8jJytLT1NXW19jZ2uHi4+Tl5ufo6erx8vP09fb3+Pn6/8QAHwEAAwEBAQEBAQEBAQAAAAAAAAECAwQFBgcICQoL/8QAtREAAgECBAQDBAcFBAQAAQJ3AAECAxEEBSExBhJBUQdhcRMiMoEIFEKRobHBCSMzUvAVYnLRChYkNOEl8RcYGRomJygpKjU2Nzg5OkNERUZHSElKU1RVVldYWVpjZGVmZ2hpanN0dXZ3eHl6goOEhYaHiImKkpOUlZaXmJmaoqOkpaanqKmqsrO0tba3uLm6wsPExcbHyMnK0tPU1dbX2Nna4uPk5ebn6Onq8vP09fb3+Pn6/9oADAMBAAIRAxEAPwD3+iiigAooooAKKK5nxr4xtfB+lLcSr5t1NlbaHn94wxnJ7DBoDc6as3UfEOj6TEJL/U7S3QkqDJKBkjqMV8/X3jvxLqc9zNLqc9vDMxb7PA5VEGMYHfGP15rnEgVn811BkbnJ5xTsaql3PrKOeGV2SOVHZQCyqwJGemakr5T0y+v9O85rC9ntZpQySPE5UsPQ16P8OPiVLFcW/h3X3zubZa3jtgdOEbPcnofeiwpU2lc9looopGYUUUUAFFFFABRRRQAUUUUAVdQ1C20qwlvbyTyreIAyOQTtGcZ4+teBfEvxHD4h8UZtJxNY2cflQMpyrMeWYfoM+1ez+OYPtPgfWIcOWe2YKExkt2HPvivmEb4lEU0bRurYIYYNHMr26mlNdSeVsxxj++QPwqYGp9Ht4rnSLq9lTzGthsRM9G7n9afJYsEmMLF/JUs5OApwMkDnP+NZ+1jdrsdCXUoq227kHrhqSYCYnaSroeGU4IPYitK7soP7KstUi3IzYilQ9DkEhqzoIpXMjRxSODJ/AhP8qcKsZK4mmtD6G8C+NLXxHotnHcXMQ1YKyTwjgll6tj0Iwfxrr68N+EelTw+K3vrqGeAeSyQqyYDk9SfQAD8c17lTjJS1TOaas7BRRRVEhRRRQAUUUUAFFFFAGR4ps7vUPDGoW1jKYrpoiY2AzkjnH44x+NfPdt40hcCPU7FJB0LooP5g19Nmvl/x/wCHE0zW726sUCWnnEPCBjyiT2H93+VY1aUKnxI1pSa2OisJNFvQ8umNCrv/AKyNfl3/AO8v9aWTR7WVubdJNuPlfqB9e9chc+DdVtEjubRluEZQ6NEdrYIz0quNa1uxV4JpJwWG394p3D6GuOWEmneEjpVRdTu7tNKS28vUXi8sMH2bscjoABVCfxpYWEYg0+zZlH3eNi/41zWmaHrOryiVbZ44j1mn4z+fJputaAbO9twLsSmRtgPTDZ/+vVQwUftu4Sqdjr/B+u6v4m8b6daxoqwwyieZUPRFOSST+H519AV5V8JPDsGjXmoyEGW5aNQZWXoMnIHpnjj2r1Wu2FOMFaKsctRty1CiiirMwooooAKKKKACiiigArjfiB4Rj1/QL17YLHeiIsrdA5XkZ9+OtdlXKeL/ABcuhW5hs/LlviD8rAkJxwT/AIVE5RiryKim3oeMZ123s7HUdJuPttq8KrLbg7l44ynf8vSun0qe5vLPzb2xEMgPC5DZFcV4Q8WJp8b6fqmVj8xmWUDhCSSQR6ZzXoUF/ZSxb4LmGRTz8rg1LubpNHKXGo+J76ZzHYfY7JSfkK5mce3YE1JY+GYNX8VaTpMivEWha4cBuQ+d2Cfw7Vs6l4g0/T7dpZrhDgcKhyzH0ArlPDHimdfGsWvzxn7PDuVYlxuKkEYyfrVQpuUriqSUUfQ+laTBpNsY4uXbmRz1Y1oVU0zU7TV7CK9sphLDIMgjqPYjsRVurOe9wooooAKKKKACiiigAoopskiRRtI7BUQFmJ7AUAY3ibxRZeFrAXl6HZM5ZY+WC9yB37V4+3iCy8WXlxeWqeVukJaFjkrk9ffNQfEDxG2r6s5G1rbYY3i35ZV9wOlcp4N0G+j1K71IrcLY2Fs83nKQqSt0SMsQR8xODjkY7da45x9vGyfodEPc1LWs+HTNfyT2eFkc7mjbgN6ke9c3On2KUrKGhfoR93NdnpesreWrLqcf2e6iPzHOAfcH0qHW9Ztb20NpFEJSesm3O36Z71lRqVoy9m1c2lKPLc5SKJrrHGI/U9604/8AR1WFUKgdCRjNCJtAZgAo5A/xqO/n1KOxMUUit50qyCJ+W4BGR6DB/SvTheGvc4qk+c6DQtb1HRrrz7C6eF+NwByrexHQ16l4G+Jf/CS6o2j31oIrwKzJLFykgXrkdVP6V4fbXbCNjIjRuo+ZW4IrpvhZqY0rxO2oTEiB0MUh9mOc/gQDVV5xSTYqcW7n0dRTY3WWNZEYMjAMpHQg06swCiiigAooooAK4P4m+J30PTLexjt2d78spcHAVRjOPfkY+ld5XDa+tzrOsTx20IaKyQh3Y4Vccn8f8KmaurFQV2eY+Fvh9qOs68FlV002Nw0tyRgOOu0epPT2r17WfBsNz4V/sLR3j0633h9uwsGwc4POeuDn2q/4ThMPh22zn59zjIxwWNY/xI1+80Hw6jWJ2TXMnk+aDygwSSPfinTjqrbhN7o+fNS+3aVrV3Z3BVzbTtEyMvynBx+Va6aKb62jvNP27ZF3eWxx+RrNuFilaW7vg0oT947Eks5z0z3ya0YfFNxaXjG5HmwHBTy4wpUY4AA4x7VhiVVi709zajOLVmRXdlPplg9zOFeUY2LuyEz3PrWLY3Ky3o+0uxaQ/fClsfgOcVf1fX7zUEdY7byrXGXDcs655HtTYrOKBFaDmN1DK3dhWmF9rvU3IreztojQ8d+HLnwvJpgluEnN/bEkxqdqbW4AOBnIarY8H6t4eksNQv48adcRLJ5sZzjK7ipU87h9K6vwfo8/jPw1c6JeXQa2tLyCdBLljGvzbgv1x0PHWvSfGFoJPDrbY9wgYNgDOFxg/oauquZNMim0mjH+GWvjVNImsmS4DWr/ALtpiTujPTB9vT6V3VctpVo+havFbsQYLmPCMq4G4YPPvz+tdTSjtYJ2voFFFFUSFFFFABTBDEocLGgDnL4UfMff1p9FAAAAAAMAdAK4L4tQQP4VhlmkZGS6RUOPlywI+b0Hv613tY3irw9D4o8O3WkzSGLzQCkgGdjA5Bx36U07bAfMWvE23lWBILuRJLg5AH8I/r+VXXRA7bRgKcD8KqeKPC+q+G9bezv1WR1AKSRNuVl7H1H0Ndr4A8HXfi9pLy98q206NirmNj5rtjoASQByOSPpRCaW71CUX0OSI+UjaCGBU5HY5/xqt4bm8+R9KlZVcFmhZzgAjqp/nXpnjP4Z32k2RvvDuy8giQtPBcn94AMkspGARjt1+teT2Gm32r33mW8aKZHGSr4C5745P5ZpVJre9gime7/B2/s59I1G1g+aaK43vIBwykYXB/4Cf8mvSq5LwB4Ki8GaRJC0wnvLhg08q5CnGdoAPYZ/WutoYCYpaKKACiiigAooooAKKKKACiiigD50+LmuQ6h4znt4Rbyx2saw+Zg7g4yWGQexOK9O+D9u0HgGF2jKGaeSQErjeM4B9xxXFeKfhb4j1bxdf3ltHE9tc3BdZZJVBCnHUdeP6V6/4d0n+wvDthpfmCQ2sIjLqu0MR1OKlQSdynJtWL1zbx3VrLbyjMcqFHHqCMGvkuWM6R4llQNHmyuWVXFvtY7WwDgEEdK+ua8V8bfC/WtS8V3upaZbwy2104k2iUKytgbshsd8nj1olFMIyaPYNNvodT0y2vrdw8M8ayKw7gjNWqw/B+kT6F4T0/Tbkr58KEPtORksT/WtyqJCiiigAooooASiiloASilooASilooASilooASlpkkscMbSSuqIoyWY4A/GuL1n4peH9LZo7Z31CUf8++NgPux/pmgaTex21LXj83xnu85j0eBVPTfOSf5Cov8AhdGo9tItf+/jf4UF+zkeyUV4rb/GbWE1BGvNOtTZZJdYlIkxg4AJbHXHOK9J8J+MdP8AF9rPLZRzRSW7BZYpgMjIyCCCQQefyosS4NbnQ0UUtBIlLSUtAHlHjjxX4lg8YvoGh3sEDNArqJQozkZOGIPPBri9L8S+LLjxK9hJr16NQ8wxnc4ECkDP3SOc89h2rufHNx4cbxhEl/JCl/bxoyMTtZTyRzWI3hHStV1aTU4dXkSeUDIRlxkAYI9+K8+ri1FuOzOmEVZM7bSW1OwTfcarcXkzD5jKfl/Begrbh14qcXEQI/vJ/hXN2OmXtvGqPqbTqOhZBn86v/Zcj5pGNcCxFaLupBKMWzq7a7gu498EquO+DyPqKmrgGto9NvP7Qt5GjuFBG/PUdwR3Fb9n4otr7wveatC6M1nHJ5yg8K6Lkjn8D+NephsUq3utWZlOFtUeUfFjxSdU8QDRraU/ZLHiUA8PL3z646fnXAh2eeO3hAMjcknoo9aoC6kuZpbr5pnZiSe7MT/jWpamPT7d553XzWG5z/QV1Nm0FZFmSCO3Xltz92PU1mXNzjhahtbqW+iadyfnYkD0FTfZS/Jq0gbvsJbI08LMwZpHISFfVvX6CvpjwNo9vong/TraDDO8Qkmk7vIwyxP48e2MV806KLm61tN2VtoMeWo6HJ6/WvfvAestNqWoaS8u5Y40mhX0B4b8M7fzpMznrE7qlpO9LSMRKWkpaAPIfiz4Ok1fXbPUYZVTdB5Tb14ypJHP/Av0rzTUND1bw9ALgTHyQcFomJC/UV9Na1pcOs6Rc2E+7bKhAZW2srdiCOhBrxvT/D+r2F5LZXt2b/TJFIJnA3qfTI6is5RT3NqbVjgofG3iDTsGC5aRR2JzV9fiT4nvNiRERc/MVUMT+dUfFOhNoF+Vjy9rLkxn0/2TW98PtDtdUtbiSYuk8cmAAegI4NQ8PSevKaeY99c8S3NmwuY2lRh99VwR9cVDoGuJo/hrxPZXpLQX1m6n5ckSEFVP055/D0roru6j024uLVzkwruJA6jGaxfhn4bHjv8A4SAX/mx2boqrKmMq5cNgZ6/KMfQ06dGEHeITkrHDWM0Qt9sW0Nt+UH1p2naPJdnz9QlYgc+Xnr9ao+ItEvPCPii80a4JL274RgOHQ8qw+o/rVqyvrpAACB6Bqpxa2CMlK1yxbOdjsiYUucADoKJ7qdYyIoyXI6VdS7vnGFjtj9a1dMlggnFxqs9uVXlYox396vmYzH03+2bSye9ksWjtFwN5HVj0r1v4M2k1w+p6xOzEkLAuR1/iPP5VyGueLdN1TRpNMtPmllZQuB0wQa9l8CaQdF8H2Ns67ZXXzpAeu5ucfgMD8Kad0ZSdonSUUlKKDESiiloAy/EcGo3XhzUINJkEeoSQssDFtuG+vavFLP4jWxt2g1ZWtr6IlJQRwWHB+h9q9/rldd+HXhbxJf8A27UtLV7kgBpI5GjL/wC9tIz9TSaLjKx4N4t8U2Gq2UdvbOJGVtxb0rJ8O+K00DWRdSea0LRbHWLByexwaveNvD2l2Hia6stDgMVjbnymeRmclx97kn1/lWHBocbdy59ScD8qUPe0RpKSS1PQYPEaeLIJNP0eAS6nqDsiI42lBjqx7ADJzXsvgvwrbeDvDcGlwEPIPnnmAx5kh6n6dAPYCvH/AITWsOj+MoXkYKJ4nhH+8cEfyxX0DVuLjuZSnzbHH+OfAFh4xihnYRxajb8RTlcgr3Rvb+RrxjxZ4L1HRIGFxYuioMrPGNyH/gQ/rivpekIBGCMg0gUmj4wjllZCQ3SoxfTRSjzRuTuQK90+LvgB7mJvEmiwRiWFCb6FBgyIP+Wg9WAzn1H0rxKBN0pIXcWU7R70WNFK53vw/wDC1rr3ieOa3G62CrJJkcKBjd+Z4H1r6PrgvhV4Vk8O+HftF0my5uwrbCMFEHTPuc5x9K72gibuwoFFFBAUtJRQAtFFFAHy/fTedrl3JcvJ5Ul07OHfO35jwTVFYhM+beUeWT8rCu/+LuiNZ61FqQBNtdrt6/ccdQB2yOfzry+1mNncGMH9y5yP9k+lbU7EyvY6G0F1ZzxTxXsgljYOh4IBByOK+ifDutQ6/osF7Gy+YVCzIv8ABJjkf57Yr5wjuEx8zYra0HxZqWiTsmlTE+djfHs3g4749aqootExufRVFeW+GPifd3HiG10fWLeMi8JWG5iG3D/3WHoeOa9SrCUXF2ZUZKSuhGUMpVgCCMEHvWTpvhbQdHk83T9Is7eTJPmJENwz1weoHtWvRSKCkpaKAEopaKAEooooAWiiigDwz4xaw13r66dHuKWMOXU9C7Ddn8tv615VLDcwS7H2tnBw3BGea9e+MOjxWGox6vFuzqCGOUE8b1UAEd+R/KuH1qxC6lIrL0Vf5CrUVKVl2HzNR1Objtb7UL2G3tgXl3YaPPTuMmvSNN8LeJWtlhjFlZRkfM68tXO2EFnpmmtqwucahFMP3bHhkx0rqX+IFzZ/uvs0AO0EMWPI9a83EVKqnyo6qaXLoYNxZy6fqdxBcTKxt5CodyBuweor3nwddTXvhSwnuH3uY9u/dneASAc/QV8u6xc3V9r889xlppyHA/3gCMfnX1holqbHQtPtGTY0NtHGy+hCgGu9VJTglI4vYxhNtF6iiigoKKKKACikpaAEpRSUtABTJjIIJDCFaUKdgY4BOOM0+igD5w13RfiBrupSz32n6nMqSGVY2B8tMf3RnHQY461majrdvqrJMwEFwo2yIxxzmvp6eeK2geeeVIooxud3YBVHqSa+cPiP/wAIvceLJJtO1GWWS6lDzOkYeCMEAEqwOW55OPf0pJ8r5h7qxjEDVFW1s0aeRztGxcjPpV6W5mhBtLqwWWaH5FLjBAHAzjrV3SLCLwvOurDUcQxDdCEODK2Oh7YOa9A8PS/b9HtLrULfbdvGGYyx4LejDI6Ec1M+Wbu9S6d1sM8C/D2wv0s/FGsp5926AxW2AIowpIUkd26deBXqlZ2iYGmIBGUUE4z/ABAnOR+daNUtiJbhRRRTEFFFFACUtIaWgBKWkpaACiiigDy74xW2qXdvp8Nutw1k27zFiyVZ+MbgPbpn3rjPD/wy1bXI0aa3Npbj5lnuFxkH0Xqa+hKKTjfcadjkNC+HGgaL5cjwG+uUXasl18wX/dT7q/zroZtIsbi5W4lg3uqhVBY7QB7dKvUUWQXYgUKoAAAAwAO1LRRTEFFFFABRRRQAUUUUAJRRRQAUUUUAFFc/4z8Rt4W8OyajHAZ5S6xRJjILN0z+Vee6b4m+KFy1zL9m0cI7DYJ3OEx127f656VEpxh8TKUWz2KiuO8P+JNfeRYvEGn2MK4Obi0nZue3yFf612COsiB0YMp6EURqRn8LE00LRRRViCiiigAooooAKKKKACiiigAoJwCT0FFFAHJeII7PxNYi2kL/AGcMHVkbDbh0YVVhtWtohGjAqBgZrzHWtH8UeFfFd9b6bdk6eZTLbw+YeI2OQBn06fhUi/ETUIG8q6tyki8EMK8ytSq813qdMI3Wh6JdSzRRkrEzkdlIqTw34hmi1BbO+jEEM52x72GQ/bp2PT8q84k+I7FeYxmub1Lxs11eKxjYEYw3pWdOFWMlJIv2aejPpDXfEWm+HLRbjUZygc7Y40Us7n0AH/6q831P4wXbMV0zS44l/v3LbmP4Dp+ZrivFnjB9a1M3Dnc4QIijpGvp/U+9c5C9zeFmRTtXlm6AfU17FjFQS3O5n+J/iqWVXS6ghA/gjhXB+ucmrEHxZ8Sw7vNFnNnpvhxj8iK8zmv44yVVzIw/u9Pzqu2p3G07YolHq5JplWR6q/xd1/zFciyRFOWURHDD3yc/lXUaT8bPDV6UjvfMtZCPmZR5iA/Uc/pXzt9peWb57lHDH/VgcfSpzZBLuIwKEYrklfftSbJcU9j680nW9N122e40y7S5iRtjMuRg9cEGtCvOPg3qRufC8+nvCEls5uXH8YfkE+4wR9AK9HpGTVnYKKKKBBRRRQBxHxGtBHp0OqIn7yA7XYdSvUD+f51474t1qzit4FMQN3IAcd8etfRmraZDq+mT2M5ISVcbgMkH1r5s8e2UNhrFxptw8N19nJW1ubaRXePAHySKOR7+lK12awlZWOZe9BTLbVbHQds9KrtcmS+iSJdxH6e9VDA27GDknNbuj6OBBdX093DHIiARW5J8x+Rk9Mcemc+1N2RSk3oNuYktLczStwOp6k1Faf2lrbraW0bRWwP3O7e5qe1vYb+bDJvwcAEfKv8A9eu20670zR7L5Nu8jLHuTVWKsZlp4ECoDcShT6LzTr7w3ptnaSth3YKSNx4zWtHry3ERlPAPQe1YGp6qbl/IQ/fOKBpdzAn8OeTaRTRNlmAY+1P8loY0lbqCM107KBAqdgKyNQjH2Zl9aGU0j0f4QXO3xDeQA4WW13Yz1KsO34mvZa+d/hffiHxtp3BbzQ0eAem5etfRFSc1TcKKKKCAooqO4nitbaW4mcJFEpd2PYDk0AYvjLVTpHhW+uUcLO0ZjhHcu3AA9+p/CvAG0Sx1W2a4jLxzrxIp6g+9dR4t8UyeJdS8yPfHZRDEMTH82Pua5CfURYztjrIm0/nXTGPKtTNu70MKayWGQpk5Brd8Nx2pvEF0RhzgbunFZEnmXEu4IeT34qG5mVFW3c4+bBcH1rnqtW5UbU99Tv8AxPo+nw6cb2zSONl67BjNeeQTSXk23cdoNTRa5dDT5tPuHZivy896XSbV44t7rgmlFNKzN76l93aOPap4qC2jeS43ntVllzU0EZRAQKoaLhchOay9Q86aG48iNnEMfmSFRnYuQMn2yR+dW5JSRgV6l8I/DkH9mXur3DLM13utmhZQVCAjIOeufT0pXCcrI4n4OadNd+NYLlFPk2ULSO2OBlSoH45P5V9E1naRoemaDatbaXZxWsTOXZUHUn3/AJelaFI5pO7CiiigQVV1Kzh1HS7qyuRmCeJo3GccEYNWq53xrrB0fw9K8cqpcSkJGGGd3OWH5Z/OpnJRi5MaV3Y8HvLM6T5kEjSQzR9Ype49Qe/4VzUl9LPeefFCXVEOQfqB/WvR/GOp6ZrOhKyMhuGACR/xhq4HT4Wms7/yD86ogjI/3qzp1nOnqaSguZaFKe4vSN7xiJfVjVFCk0uZgZUBBZckbhnpVryr2+mEciyM2cbcVp3Hhy6gSA24UOxxJv8A4e+fyqoq2iG7JXYzTbeF3kfmTa21XdcFlHAJHrjFaUhVF9KhWzuLBGkciROMsOPbpWho/h7VfFEl1DpqxGWCHzfLd8F+QMA9Aee/pWzTW4RqwaumZYlzIF7mrarKy4AwKktvDHiL7f5DaJfeep2lfIb+fT8a7HS/hz4h1AK00KWMRGd07fN/3yOfzxSK5kcRKVhADHJJ5r2P4Q3Rl0C+t9o2xXAYNnruUf4frSTfCHR59FNq91N9vLBhegfd9gmcY/X3rq/DPhqy8LaV9hs2kky2+SWTG52xjt29qCJTTVjZooopGQUUUUAFeM/Ey/1k30V19nMunohje2HJQ56n1P09K9mrlr3w1dTzoBJHJD5wYl2IbbkZzxyetTJXViotJnlOteENQtfBtr4hit5WLo5khb78CPja54z0Bz6bh71xfh0y2+pyWw6zxssZPI39V/WvdfH2leJLqVJ9LuZmswuxoLdijISMEnH3h/KvGJrW207TzdTzOssEwUwrgOCGHAPY49qSikuVIJTejZp2PjnTLf5dQ0mSOdeGaIBhn8cGn6hrdrqjwXunsDESySwSqAysMYOAc4PrXfx/C7wl4phh1i0vb1ra5USDy5EAY98/Lwc9R2NeYeOdFXwfrtvaWgIhQOqb+Sy5B5PfrVwtGSbFWj7SDjEq6o2pS4XaGg7LEOPx7muk+EN9LD8RjAzmNbm2ZXRv4mADfnwT+dca2uXTx7Iwie45P4Zrrvg/Zm78fx3DBn+zwSSlvQkbcn/vo1vVnGWkThwtGpTd5n0TRRRWB2hRRRQAUUUUAFFFFABRRRQAVx/jD4caL4wXzJ1a0vR0uoAAzezDo34812FFAHJeCPAkPgiG5ig1S8u0nIPlykBFPqFHf3ql8RPh4vjO0SW1uFt9RgO6NpB8j8Y2tjkduefpXdUUDufMMvwv8a294tt/YrSbjgSxzIU+uc8D64r174ZeArjwnb3F7qTqdRuVCGKM5WJAc4z3JNegUUCCiiigAooooAKKKKACiiigAooooAKKKKACiiigAooooAKKKKACigH/2Q==')))
im.save('1.jpg', 'jpeg')
im = Image.open(BytesIO(base64.b64decode(b'/9j/4AAQSkZJRgABAQAAAQABAAD/2wBDAAgGBgcGBQgHBwcJCQgKDBQNDAsLDBkSEw8UHRofHh0aHBwgJC4nICIsIxwcKDcpLDAxNDQ0Hyc5PTgyPC4zNDL/2wBDAQkJCQwLDBgNDRgyIRwhMjIyMjIyMjIyMjIyMjIyMjIyMjIyMjIyMjIyMjIyMjIyMjIyMjIyMjIyMjIyMjIyMjL/wAARCAGAAIADASIAAhEBAxEB/8QAHwAAAQUBAQEBAQEAAAAAAAAAAAECAwQFBgcICQoL/8QAtRAAAgEDAwIEAwUFBAQAAAF9AQIDAAQRBRIhMUEGE1FhByJxFDKBkaEII0KxwRVS0fAkM2JyggkKFhcYGRolJicoKSo0NTY3ODk6Q0RFRkdISUpTVFVWV1hZWmNkZWZnaGlqc3R1dnd4eXqDhIWGh4iJipKTlJWWl5iZmqKjpKWmp6ipqrKztLW2t7i5usLDxMXGx8jJytLT1NXW19jZ2uHi4+Tl5ufo6erx8vP09fb3+Pn6/8QAHwEAAwEBAQEBAQEBAQAAAAAAAAECAwQFBgcICQoL/8QAtREAAgECBAQDBAcFBAQAAQJ3AAECAxEEBSExBhJBUQdhcRMiMoEIFEKRobHBCSMzUvAVYnLRChYkNOEl8RcYGRomJygpKjU2Nzg5OkNERUZHSElKU1RVVldYWVpjZGVmZ2hpanN0dXZ3eHl6goOEhYaHiImKkpOUlZaXmJmaoqOkpaanqKmqsrO0tba3uLm6wsPExcbHyMnK0tPU1dbX2Nna4uPk5ebn6Onq8vP09fb3+Pn6/9oADAMBAAIRAxEAPwD3+iiigBKKWigAxRRRQAYoxRVa+1Gy02Dzr66ht4+m6Vwo/WgCzSUK6uiupBVhkEdxS0AJRRRQAUUUUAGKWkpaAEpaSigBaKSigBaSiuV8a+L7Hw5ptzbvM6ahLas1sqgjcxyowe2DzQByfjr4oXGm6umn+H2R5baRlvGljym4cbfw56e1eV6lqeo+IL17zUrp5mY5Ck8D6DtVHe2yRmcs55ZmOSxPcn1zU8fCAe1M6IwSOg0Dx/4k0LSxBBefaI/MDkXGXK4Iyqk9FIGMflive/DXiG08T6JBqNowG4YljzkxPjlT9P1GDXy/C4QSIf4XxXY/DvxOnhfxHILl2Gn3ibZQq7sMM7WA/Ej8aGTKOl0fQtFRWtzFeWkNzCxaKZA6EjGQRkcGpaRiFFFFABS0lFABRRRQAUUUUAFeM/GuxubzVNMNnEryRwsHw/zAFuMjPTj+dez15D8b9Fm+zaf4hs28uW3Jt5mV8MVJyv1wd351Mr293cqO+p5rp2i3H9pRJqEJS2aQM7g8cDpntzU9zp6RC5PmiO4ExSC0Ckswzgc+/rVHT/FF7bqBcgTx9MsMH862E8W6cCjGCTchyoIBCn29K4ZSxCle33HUnGxl6zYG2FvLDG/m3fytCFzsZMA/rmpYNF1C6mAS2ZQBgFzin3fi52YpY243sScsMkk+1ZUmoa3e3P2eSa5DnA8pcr16DA9a0p+3cbOy9QfLc+o/Dk7XHh2wd1jVxCEYRtkArx/StSsrwzpI0Lw3YaaCCYIgGYLjLHknH1JrVrrV0tTke+gUUUUxBRQKWgBKWkooAKKKKAFrO1zSYta0qaxmClXHRhkVoU2SRIo2kkYKijJJPAFAHz2+k6roN5rWnW9hFdi1dG8iT+MOOMevANLpeiaZrlu9xP4fezlRtrRsWUZxnjpkfhV7x54tXSviCl/ZB57O5tFjnRjjcyMeV9CAR9cmrln4w0i8hLW9wAccpJ8rL+BrKPJL3os6VzNanOW97cQXQtdH8NCLdKI0lmQoSD/EQRnHBPWt7wp4eutV8S6jepbFwk+xZ2GFUrgHn1qjrHjy1s4H+ykXE+MKF5APua6T4TeLbeHTTpOqSrFcySGVJmOFkZuoJ7H+dXGGtyZuyPWUXZGq5zgAZ9aWl6jikqjAKKKKAClpKKACilooAKSlpKAFrxr4mfES4tdRg03TfKe2Ulp3V93mjpt/2SDn17V2/wAQfEzeHvD0rW5RruUbVUtyFPG7HpnivF9B8MSeN9XMNtG9umd00yruSH8+uRkAeuPQ1hUd3yW06mkF1ZZvdOg8RaWlwjExt80cg6of8exFcrf6FqFgMyW6yxZwJY+n4jtXQeIbeLw54jmg0eeextkk+zNDI25ZSgGXYHuxJ59MVUvry9u0SOfEcJ5BUcNXPTpVac1GLXKb88XHXcxIrXbgvjcOijoKv204CrjIDKGGRjIPQ0CMQoZpWVYxx161BZ2ckhNxGHlSPdIwj5/dAFmA56KOfpXpQaWxyyblud14V8f65puu6fpSsLyzuGWIQSkArlgMq3bGeh4r3qvkSzv2XX7aZuTEyvszjoc4r6m8P61HremrcAKkw4kjDZwfX6GonJe0cQt7qZq0UUUCCiiigAozS0YoATNLSYooA8r8daWuu+InlW4k8q3jEThQMd8g/if516Lo2kWWiaXDY2FusMKKOF6k9yT3PvU4sLTypIvs8eyR/MdcfebOcn8as1KjZ3Kcrqx87/FC7g1bxfcJDEkYhIhLqOZHHBY+/b6Cs3SrqzWF9JvQIkQ4V3frz1z2NVPElykGv6rPBIZEiu5EiZuu7eQM+4wT+FZ6wrJYQmQbj5fVuepNKpQVVWvYcKvIaPiGbTdTK2dltzCwKlOQQAeAfU8n8K0Ph/qP9jaq9nIim21FTau+wF4S/wAu9c/UZHQge1cxeILa33wrtdCHUr2Ib/Ct3S7yzi1PSr2WKSSGeZN0UX3sgjIH0q6dFUo8opVHN3PR/Bvw2tIfDNxPrunpLfi6lmgd+GAA2rnHUEruweORWh4E0YeHtYdTLJsvo2MfmDO/ByPm/wC+q9GIBGD0NV1sbZUgQRAi3/1WeSv40mru4J2VixRRRTJCiiigBaKKKACiiigQUUUUDPEvjD4ItbeT/hI7ScQNczKlzAQdrvg4cY6HA5/Osz4Y+DYfFP2yTU7lms7VViSGJtrFjzknGcAD8T9K2/jdrFzFcadpkJaOMIZ2cEjcTkY/T9a6j4QWH2TwNFcMrebdzPKxYYJGdo+owP1qbu+49LHF/E/wBb6Ho8eoaRcLBblvJniuMvkt0IbB29x+Iqt8JvB0Opal/aV3dBksHSWKGI5DP6kkdMgdK9X8eWcl94H1eGIDzRbmRM54KfMCMc544968k+C2pXMXjGazdmdLq3bfuOcFeQf5j8aG5cy10GkrHv1JS0VRIlFFFABS0lLQAUUUUAJRmlooAKKKKAM7U9B0rWTGdSsILkxghDIuSufSrsMMdvCkMMaxxRqFRFGAoHQCpKKAEZQylWAKkYII4NYuk+EtD0O/mvdM09LeeZdjlScYznABOByO1bdFAXEopaKAEooxRQAUUUUAFFFFABRRRQAUUuKKAEooyKglvbSB9kt1DG/915AD+tAE9FV49QspZBHHd27uTgKsqkn8M1YoAKKKKACiiigAooooAKKKKACiiigBssqQQvLKwWNFLMx6ADqa8N8UfFTVdTuJbTSHNlaDjzF/1rD1J/h/D869C+I2otDoEmnxOVe5Ri7D+FFGSPxwR+dfPsZHlNM3AdiRSjJNtdjanDqx7NNLIXlu5yxOSzSEmqty8CjfLczHP8R5z+dMmu1z8qlyOw6fnQtqxha5uMGQ/dHZR7VdzUrwaklldx3VrNcpcQuHjkB2lWHQ9K+rfCWq3GueE9M1O7VFnuYA7hAQM/jXyzEY3hw6g545FfUng50k8GaM0SsqfZIwFY5IwoHWkzGojbooopGQUUUUAFFFFABRRRQAUUUUAeS/FC6sE1xLbULtrUy2oEbKfvKSwP5c/mK83uNHsr2eNNO1a3lRVwsDttJ/HvXs3xQ8PaRqeiJqOpwOfsZwZ4s7okbvgdRnFeD6n4eSGy/tLS7tL2zBwzKfmT61zujJScoy3OiErxsaX/CLauh3C0jbHT94P0FRNoer3MgtntjDnrI5+UCsyx8S6tYKEhu3Kdlf5h+tdfpqeKNZjE0jw2kRGVMkR3P9Fz0+tZv6wtrFXRyGoafPpVyLeYfL1SQdGFfTXgZWXwLoocEN9lTgjB6V4DfrfX3iS38MXEcc80ssYWVCRjdjt24r6bt4I7W2it4l2xRIEQZzgAYFdFKU3G01qY1Gr6D6KKK0MwooooAKKKKACiiigAooooAgvbdbqymgYZDoRj1rx3UNJ07S5r6WO0S3aWMrOoG0HGeo6V7TXjnxzs7lBpeoWxkIYNDLGp4YDkHH5/nSk0tWaU3ZnDeD20qy1CeS/jQkLuhkZd20g84Hr7+1epW7QTwq9uyyK/Rl5zXz3Je6mHHkxmNuxIxXpnwY1i6sfE0tpq210vkAglbjy5B0A7fMDj1yBTXkVNnonhLwMbTxRfeJ9SXFzJ+7tYSP9UgABY+5xx6D68d9S0lBjuFFFFABRRRQAUUUUAFFFFABRRRQAV5T8ZBdudLVAogAkJOeS3H8v616tXnnxTu4WsINPLxrMVa4BbqFHB/r+QqZuKV5bDje+h4tbQJ9py+C2K1UzEyvGxVlIKspwQexFc7YXlu127fOkjfe3n09KmvtaeOcW9ogkk7nrXTGUUiJJtn0r4Q8QL4h0KKdmH2qL93cL/tDv9D1/wD1VvV8w+DfGWpeGPE8Nzcsj282IbiE8fISORjuOo/Ed6+ns5AIrF2voVZrcKKKKQBRRRQAUUUUAFFFFABRRRQAV4X8WoL4eKxcXULC0ZUS2kx8rKACwz65LcV7pXn3xikgj8E5cx+cLhDGGxu9Dt79+cU0wPDjaWsWplrpP3fJyOMGsvSbOS51eG2EhhimfBYfex9a29VuYFk2NkvtDHHvWSk0X2qNxI0YQ7iw6j6UVoK0mnqOnJ3SZ614Q8K6DY+LdPkeEySAsYzK5b59pxx09a9nrxHw/f8AhnT9esrptTvL6VZQqbwcKT8ucAds17dXNhlJQtJ3ZdW19AoooroMwooooAKKKKACiiigAooooAK5nxV4H03xc8El7NcwyQgqrQsOQexBBFdNR0GaAPn7x98LL7TZ477T5zPYKio7smXjOcfMF6jpyBWJ4C0Kym8Rxpdo08oUvArJlWlGCoYf3f8ACr/jrxtq3iy9n06DzItOSTMVui/OxXozEcn1x0rc+GngPXJb2z1bVGnt7KFhKiu5EkpHQY6hc+vaod2y1ojpr7RI9F1RZZILd5mIlDIMAnPP45r0pW3orDuM1natosGrmAyu6GFsgp1IPUVoRoI40jXO1QFGfQU4xswlLmSHUUUVRAUUUUAFFFFABRRRQAUUUYoAKKWigCEWtuJvOFvEJf7+wbvTrUtLRQAlFLikxQAUUUYoAKKMUUAFFFFABRRRQAUUUUAFFFFABRVe51CysmRbq7ggL52iWQLux6ZrBb4heFk1Q2DavAsmB+8J/dk+m/pQOzZ01FQwXltdAG3uIpgRuBjcNx68VNQIKKKKACiiigAooooAKKKKACiiigArjviN4ubwpoSNbMFvLlykTEA7ABlmx+Q/GuvlcRRPIQSEUsQPavnL4g6zLr3ip5b2IRx26BLeMkkbOu73JP8AL2pcyvYuEbs5a81ye/upLm58+5lc5aSRiSaga/tJcR3MciA9weR9KSS4EjkJ90d+1JI1vLFskQMPWq3NtBtrqNzp00r2V/OEKkKysVYZ68ivoz4a+Nl8WaJsuWgTULbCvGjcugAAfB98g/8A16+X5E+zuyoS0bcjPUV7p8CNBvrawvNZnEa2tz+7hUqC7bTy2eqjqMd/woexjI9jooopEBRRRQAUtJRQAUUUUAFFFFAAQCCDyD1r5+1q8tLDXbnR/ENjExt5mWKYplShOVJ9MqR7V9BVyPiW0ij1FbgRKGnT5n2/eI46/TFZ1KamtS6crM+d/EdiljefaLMKdPnOUMfKofT6elY5au/+I6Q6bHBPBCipcMY54wMB+OGx6j1rzmPaQCWJGOBTpc0VyyLlvoTxw+c4/u96+mPhQpX4fWQIwPMkxx1G81846dBcX17DaWkLSzysEjjUcsT2r6x8P6WNE8P2Gmg7jbwqjH1bv+ua0ZnI0qKKKRIUUUUAFFFFABRRRQAUUUUAFVr+xi1C0a3lyAeQy9VPqKs1xnxB8TNpGnrYWr7bu6U5YdUToSPc9B+NVGLk7ITdlc8U+JWqWsevzaFfJNOtmw23VuQockA9D6ZwfcGuID2R2rCbqQ9NrKq/rzXYanYxT2zsyjcoJBNYNhab5TJHA7qnLbFziqlRcXqylU5lc9q+HFz4W0S1t5GWztLi4/dLcTH94ZMZKlj0/TOK9YV1dQyMGUjIIOQa+S9Rtb+8083FpZTz2cThZZkX5UcjIB9OK9n+DOrTyeHm0m7lVnt/nhBPO1iSR7gH+dYNqEuS9x2bVz06iiirJCiiigAooooAKKKKACiiigAr5j8XeL7u88W313Pbo1t5pjTy33bVX5R/LP419KX93FYWE93O6pFChdmY4AxXzf4t8N28cUmp2EiIHJeSHPynPdfT6Vm8QqU0r7lqnzpmVdajFeQpHbOHab5VGcH8a6a3nsfDGkCDfGJm+aSV+rEjoBXAafZCYvglCeNw6itaLR4Fk82UvNJ/ekbNdNRVarXK7GXuQVj6A8G6bpl54BSGCMPDeiSSbcMEyMTnPuMAD6CsPwj4M1Dw54ntru8YzRvG6JxnYSAc/KMZ6jn1qb4QmYaRqCNJm3Wddkf91tvzH8fl/KvRqxlTSeuti4zdgooopiCiiigAooooAKKKKACiiigDl/Ht9PZeG28m0W682QI8T8ApySP0rwzxVbS2ml2c1tb38djeAtEJ1+RGBIKA9/Ue1e8+J5hJPYWGwlppCwOOOOP6msbx3e+HdE8JRaRrcc7w3MbJF5CBmVl/iGSMEE1HKnK7LTsjwLSpIo0w0ig+5p+q6o9rP9jixHNgFnk42g8jGfY1c8K+Hbe/8T6bbSS/u5LhCR5WQy5BPfp/jXqnxB+EcfiG8bVdF8mC8f8A18DfKsv+0COjfofat/btxtEz9mua7Oj+GWhWOkeEbe5tZhcT36ie4uA2d7dMD2XkfnXZVheDNGufD3hOw0u7kjeaBWBMZJUAsSBn2BxW7WaGFFFFABRRRQAUUUUAFFFFABRRRQAhRWYMVBZehI5FYnifwppniywW11GNsoSY5UOGQn0/T8q3KKAPH4PgzqFjq9pNa6+jWsMqufMiIcAMDgDJB/SvYKKKSVh3CiiimIKKKKACiiigAooooAKKKWgBKKXFJQAUUYpcUAJRRilxQAlFGKKACiijFABRS0XRQB//2Q==')))
im.save('2.jpg', 'jpeg')
```

---

> ### Author Response · Authors · 2023-11-18
>
> Thank you for your helpful comments! Our replies to your questions and revisions to the submission are stated below.
>
> **W1**: One notable weakness of SyncDreamer is its computational cost, particularly the significant memory requirements associated with the volume-based approach. This could pose scalability challenges when aiming for high-resolution image generation, which is an important consideration for practical applications.
>
> **A1**: We agree that the volume construction consumes relatively large GPU memory and computations and limits the scalability to higher resolutions. Currently, we can generate 4 instances (64 images) on a 40G A100 GPU using about 40s, which is slower than Zero123 using 23s to generate 64 images on the same GPU. Exploring more effective and efficient ways to correlate multiview images is indeed an interesting and challenging problem and we will continue to delve deeper.
>
> **W2**: A noticeable gap in the level of detail between images generated by Zero123 and SyncDreamer raises questions about the specifics of this discrepancy.
>
> **A2**: The generated textures are sometimes less detailed than the Zero123. The reason is that the multiview generation is more challenging, which not only needs to be consistent with the input image but also needs to be consistent with all the other generated views. Thus, the model may tend to generate large texture blocks with less detail, since it could more easily maintain multiview consistency. We have added this discussion to the limitations of Sec. 1.3 in the supplementary material.
>
> **W3**: The fact that SyncDreamer is fine-tuned on Zero123 raises concerns about its generalizability and performance when trained from scratch, especially considering that both models use the same datasets.
>
> **A3**: Finetuning from the Stable Diffusion model still produces multiview-consistent images but with worse quality than finetuning from Zero123. Based on our observations, we find that the batch size plays an important role in enhancing the stability and efficacy of learning 3D priors from a diverse dataset like Objaverse. However, due to limited GPU memories, our batch size is set to 192 which is much smaller than the 1536 used by Zero123. Therefore, we choose to rely on Zero123 to leverage its strong 3D priors trained with large batch sizes. If we use advanced training strategies like accumulating gradients or using more GPUs, finetuning from Stable Diffusion may produce improved results. We included an additional explanation in Sec. 4 of the main paper.
>
> **W4**: The concatenation of features across all views in the volume, without a mechanism to handle occluded views, might lead to suboptimal performance, particularly in cases involving heavily occluded objects.
>
> **A4**: Thank you for this suggestion. We agree that using a more advanced attention mechanism to make the feature volume occlusion-aware could be a promising technique to further improve the quality (though with some additional computation and memory consumption). We will explore this interesting idea in the future.

---

> > ### Author Response · Authors · 2023-11-18
> >
> > **Q1**: The paper chooses to use 16 views for its volume-based approach. It would be helpful to understand the rationale behind this specific choice. Is 16 views the minimal number required for effective multiview image reconstruction, or was this number selected for a different reason? Given the extra computational burden associated with more views, why not start with a smaller number, like 8 views, to assess the trade-off between computational efficiency and performance?
> >
> > **A5**: NeuS usually requires more than 50 images for accurate reconstruction, while simultaneously generating this amount of images using stable diffusion is too expensive. As a trade-off, we adopted a 16-view setting to balance accuracy and cost. Following your suggestions, we conducted further experiments and found that it is possible to generate fewer views for reconstruction. The experiments are given in Sec. 1.8 of the supplementary material where we use 4 or 8 generated views for NeuS reconstruction. We observed that using 8 views still leads to reasonable reconstruction while 4 views are not enough for a good reconstruction. Generating 8 views may further improve the generation efficiency of SyncDreamer.
> >
> > **Q2**: I tried the model and found its sensitivity to the size of objects when using the code and released models. Is this sensitivity a result of the volume-based approach or if there are specific reasons behind this behavior.
> >
> > **A6**: The reason is that changing the foreground object size corresponds to adjusting the perspective patterns of the input camera and finally affects how the model perceives the geometry of the object. The training images of SyncDreamer have a predefined intrinsic matrix and all are captured at a predefined distance to the constructed volume, which makes the model adapt to a fixed perspective pattern. Thus, our generation results are sensitive to the crop size of the images. We have added this explanation to the Sec. 1.2 of the supplementary material.
> >
> > **Q3**: Epipolar-based attention mechanisms, as demonstrated in works like Generalizable Patch-Based Neural Rendering (ECCV 2022), are technically equivalent to volume-based attention. Did the authors consider or investigate this direction, and do they have any comparative results or insights regarding the choice of attention mechanism in SyncDreamer?
> >
> > **A7**: Epipolar line attention constructs epipolar lines on every image and applies attention along epipolar lines. Our depth-wise attention is very similar to such an epipolar line attention. If we project a 3D point in the view frustum onto a neighboring view, we get a 2D sample point on the epipolar line. In epipolar line attention, we need to maintain a new tensor of size $H\times W \times D$ containing the epipolar features. This would also cost as large GPU memory as our volume-based attention. We have added this discussion to the Sec.1.11 of the supplementary material.

---

### Official Review · Reviewer_mZRr · 2023-10-30

**Soundness:** 4 excellent
**Presentation:** 2 fair
**Contribution:** 4 excellent
**Rating:** 8
**Confidence:** 5

**Summary:**

The paper presents a method for single-view novel view synthesis. It initializes from a pre-trained model Zero123, a finetuned StableDiffusion model on Objaverse for (relatively) consistent NVS. At each diffusion step, the UNet is conditioned on all target frames synchronously. This is achieved by constructing a feature volume from all the noisy intermediate target images at time step t, then projecting to the camera frustum of the desired target pose. Afterward, depth-wise attention is used to inject these features into the UNet to denoise a target view at time step t-1.

**Strengths:**

* The task of single-view novel view synthesis is extremely challenging and well-motivated.
* Evaluations were done with reasonable metrics and against SOTA methods, and a large margin of improvements can be observed. Cool renderings and additional applications are shown.
* Through ablation studies were done to justify its design choices.
* The method is novel and reasonable, the consistency achieved is significantly beyond prior methods.

**Weaknesses:**

* The biggest weakness of this paper is its writing. I feel the notations used in section 3.2 are very confusing. Following the standard notations in score-based generative model literature, for quite a while I thought the paper somehow extended the UNet so that it is able to take multiple noisy images as input, but in fact if I’m not mistaken, it still just takes a single noisy image as input, and the conditioning from the other noisy target views are just the injected features.
* I failed to understand where the features used to create the volume came from. Is it extracted with another network? Please be more specific.
* Please mention that the computation of viewpoint difference is discussed in the supplementary.
* A small introduction to zero123 could be helpful, as many of the design choices are very specific to zero123 and require some prior knowledge of how zero123 works.

**Questions:**

None

---

> ### Author Response · Authors · 2023-11-18
>
> Thank you for your helpful comments! Our replies to your questions and revisions to the submission are stated below.
>
> **Q1**: Additional explanation on the multiview diffusion model. Following the standard notations in score-based generative model literature, for quite a while I thought the paper somehow extended the UNet so that it is able to take multiple noisy images as input, but in fact, if I’m not mistaken, it still just takes a single noisy image as input, and the conditioning from the other noisy target views are just the injected features
>
> **A1**: We extend the diffusion models to process multiview images. According to the derivation in Sec. 3.1 and Sec. 3.2, we indeed require a model to take multiview images to get denoised multiview images. A direct implementation of the multiview diffusion formulation is to use $N$ different image denoisers and each one is in charge of denoising a specific one of the multiview images. This solution would be too expensive. In our task, we find that the multiview diffusion can also be implemented by a single-view denoiser to repeatedly denoise $N$ images with all other views as conditions, which is cheaper and easier to implement. Because the single-view denoiser can be pretrained from large-scale 2D datasets such as the Stable Diffusion, which improves the generalization ability of the model.
>
> **Q2**: Where the features used to create the volume come from? Is it extracted with another network?
>
> **A2**: We have added the description in Section 3.3. They are extracted by several convolution layers on the latent feature maps of these images in the Stable Diffusion model.
>
> **Q3**: Mention that the computation of viewpoint difference is discussed in the supplementary.
>
> **A3**: Thanks for the suggestion. We have added it in Section 3.2.
>
> **Q4**: A small introduction to zero123 could be helpful.
>
> **A4**: Thanks for your suggestion. We have added a description of zero123 in Section 3.3.

---

> > ### Comment · Reviewer_mZRr · 2023-11-22
> >
> > Thanks to the authors for their replies, I do not have further questions and will maintain my score.

---

> > > ### Author Response · Authors · 2023-11-22
> > >
> > > Thanks for your time and efforts in helping us improve the paper!

---

### Official Review · Reviewer_bdx1 · 2023-10-30

**Soundness:** 2 fair
**Presentation:** 3 good
**Contribution:** 3 good
**Rating:** 8
**Confidence:** 4

**Summary:**

This paper proposes SyncDreamer, which can generate a set of consistent multiview images given a single image as the condition, and the generated images can be easily reconstructed to a mesh with a NeRF-based reconstruction method (e.g., NeuS). The proposed method extends Zero123 with several additional modules to achieve multiview consistency: i) a 3D feature volume to collect global features from all viewpoints and a corresponding 3D CNN to further process the feature volume; ii) a depth-wise attention module to inject the global features from the 3D volume into the UNet of each view. SyncDreamer produces decent multiview images and can handle input images with diverse styles. Experiments show that SyncDreamer outperforms previous works in terms of both image and mesh quality.

**Strengths:**

- The paper presents a novel method that extends Zero123 to generate consistent multiview images. The designs of the 3D global volume and depth-wise attention are reasonable.


- The writing is clear and easy to follow, with nice figures to illustrate the core ideas of the method.


- The experimental results show that SyncDreamer can generate high-quality multiview images and meshes given a single condition image, consistently outperforming the baseline approaches both quantitatively and qualitatively.


- The supplementary material contains a local webpage showing many promising qualitative results, which demonstrates the capacity of the proposed method.

**Weaknesses:**

- The model seems to be limited to a fixed set of viewpoints based on the concatenation design for constructing the 3D feature volumes. In contrast, the initial diffusion model, Zero123, can generate novel-view images at any specified viewpoint. The experiments in the supplementary material also show that a new model needs to be trained to generate 16 images with a different elevation (i.e., 0$^\circ$). Although the model successfully generates diverse 3D objects in the current setting (16 images at an elevation of 30$^\circ$), this limitation still narrows the scope of application of the proposed method. Further analyses/discussions will be helpful: i) what will happen if the same model generates *another 16 images* with different elevations using one of the generated images as the condition? Will the model fail terribly, or some reasonable results can still be produced? ii) The authors might argue that a trained NeRF or NeuS could synthesize other viewpoints, but the quality of the rendered images may be lower than that of the ones generated by SyncDreamer, and some related results would be appreciated.



- For quantitative evaluations, the settings are unclear. The generation process has randomness, so how are the generation results used in quantitative evaluations determined? Is it running multiple rounds and then taking the average metrics, or taking the best numbers out of multiple rounds? The concrete settings must be clarified, and it would be better to report the mean and standard deviation if multiple rounds of generation are executed for evaluation.



- The ablation studies in Fig.7 seem to suggest that the proposed model components (3D feature volume + depth-wise attention) cannot learn proper 3D shape priors, and the method has to rely on a frozen Zero123 -- when the UNet weights are fine-tuned together with the new modules, the geometry of the generated images becomes broken. If this speculation is true, the proposed modules can only learn local consistency and must be plugged into a base model with 3D priors (e.g., Zero123) for generating plausible multiview images. This is not necessarily a flaw, but it indeed narrows down the scope of the paper, as the proposed method is more like an extension of Zero123 rather than a general solution for multiview image generation.

**Questions:**

The main questions are listed in the weaknesses section, and there are some minor questions:

- The details of the mesh reconstruction process are missing. The authors mentioned in the caption of Fig.1 that NeuS is used to reconstruct the mesh from the generated images, but there are no relevant implementation details. How long does the NeuS training take, and does it always produce reasonable reconstruction results (as the generation results are random, sometimes wrong or inconsistent images can be generated)?

- Some recent works, like [MVDiffusion](https://arxiv.org/abs/2307.01097) and  [MVDream](https://arxiv.org/abs/2308.16512), can also generate consistent multiview images while their methods directly exchange multiview feature inside UNet blocks. It would be beneficial if the authors could discuss the pros and cons of the proposed method and those relevant ones.

---

> ### Author Response · Authors · 2023-11-18
>
> Thank you for your helpful comments! Our replies to your questions and revisions to the submission are stated below.
>
> **Q1**: The model seems to be limited to a fixed set of viewpoints based on the concatenation design for constructing the 3D feature volumes. i) what will happen if the same model generates another 16 images with different elevations using one of the generated images as the condition? ii) The authors might argue that a trained NeRF or NeuS could synthesize other viewpoints, but the quality of the rendered images may be lower than that of the ones generated by SyncDreamer, and some related results would be appreciated.
>
> **A1**: We agree that it is a limitation that we can only generate images on fixed viewpoints. The choice of using fixed viewpoints is based on the following considerations: 1) Learning the multiview joint distribution on fixed viewpoints is easier than on arbitrary viewpoints. 2) A set of fixed viewpoints around the object already enables complete 3D reconstruction for most cases. We think it would be a promising future direction to generalize to arbitrary viewpoints. We have added this to the limitation in the supplementary.
>
> i) We included the re-generated results in Sec. 1.6 of the supplementary materials (Fig. 5), which are plausible but different from the original input. Since SyncDreamer is designed to generate images of the absolute elevation of 30 degrees on the object, the re-generated images are still fixed on the elevation of 30 degrees.
>
> ii) We show the novel-view renderings of the NeuS trained on the generations of SyncDreamer in Sec. 1.7 (Fig. 6) of the supplementary material. The image quality of the NeuS is reasonable but indeed blurry than the generated images of SyncDreamer.
>
> **Q2**: Clarification on the metrics computation. The generation process has randomness, so how are the generation results used in quantitative evaluations determined? Is it running multiple rounds and then taking the average metrics, or taking the best numbers out of multiple rounds? The concrete settings must be clarified, and it would be better to report the mean and standard deviation if multiple rounds of generation are executed for evaluation.
>
> **A2**: For the metrics computation in the main paper, we generate one instance and compute the metrics using this generated instance for all baseline methods and our method. To show the randomness, we generate 4 instances for an input image using SyncDreamer and report the min/max/average metrics in Table 1 of the supplementary materials. We agree that exactly evaluating the generation quality for single-view reconstruction is difficult because the generated results can be entirely plausible but not identical to the ground truth.
>
> **Q3**: Questions about finetuning setting.
> The ablation studies in Fig.7 seem to suggest that the proposed model components (3D feature volume + depth-wise attention) cannot learn proper 3D shape priors, and the method has to rely on a frozen Zero123.
>
> **A3**: Finetuning from the Stable Diffusion model still produces multiview-consistent images but with worse quality than finetuning from Zero123. Based on our observations, we find that the batch size plays an important role in enhancing the stability and efficacy of learning 3D priors from a diverse dataset like Objaverse. However, due to limited GPU memories, our batch size is set to 192 which is much smaller than the 1536 used by Zero123. Therefore, we choose to rely on Zero123 to leverage its strong 3D priors trained with large batch sizes. If we use advanced training strategies like accumulating gradients or using more GPUs, finetuning from Stable Diffusion may produce improved results. We included an additional explanation in Sec. 4 of the main paper.

---

> > ### Author Response · Authors · 2023-11-18
> >
> > **Q4**: Details of the mesh reconstruction process are missing. Does it always produce reasonable reconstruction results (as the generation results are random, sometimes wrong or inconsistent images can be generated)?
> >
> > **A4**: We train NeuS for 2k steps with 4096 rays on every step, which costs about 10 minutes. In all our examples, NeuS succeeds in reconstructing the shapes even though some minor inconsistencies exist. The reason is that the MLP-based representation of NeuS inherently has a strong regularization ability to smooth out these minor inconsistencies and produce a plausible reconstruction. We have included additional implementation details about NeuS in Sec. 1.1 of the supplementary materials.
> >
> > **Q5**: Additional discussion about concurrent works MVDiffusion and MVDream.
> >
> > **A5**: MVDiffusion utilizes a correspondence-aware attention layer based on the known geometry. In SyncDreamer, the geometry is unknown so we cannot build such one-to-one correspondence for attention. MVDream applies attention layers on all feature maps from multiview images, which also achieves promising results. Applying such an attention layer on all the feature maps of 16 images in our setting costs unaffordable GPU memory in training. Finding a suitable network design for multiview-consistent image generation would still be an interesting and challenging problem for future work. We have included an additional discussion in Section 1.11 of the supplementary material.

---

> > > ### Comment · Reviewer_bdx1 · 2023-11-21
> > > **Thanks for the response**
> > >
> > > Thank you for the detailed response. The updated paper and supplementary document resolved most of my questions. The updated supplementary provides quite informative additional experiments, which help better understand the capacity of the proposed method. I will increase the rating to 8, but there is still one thing that I do not understand:
> > >
> > > It seems that you attributed the inferior performance of "init from SD instead of Zero123" and "fine-tune the UNet" to the small batch size, which leads to worse 3d priors. I wonder if any reference in the literature related to this observation exists. I believe 192 is not a "small" batch size in common sense, and after all, training a multiview diffusion model at the scale of Zero123 is hardly affordable for most people.

---

> ### Author Response · Authors · 2023-11-21
>
> Thank you for the discussion! The batch size problem is stated in the supplementary material of Zero123 (Sec. E.1) that the batch size 192 leads to slower convergence and higher variance and the original Stable Diffusion uses a batch size of 3072. MVDream also mentions that the batch size affects the final quality.
>
> The following is the paragraph about batch size from Zero123:
> "First, we attempted a batch size of 192 while maintaining the original resolution (image dimension 512×512, latent dimension 64×64) for training. However, we discovered that this led to a slower convergence rate and higher variance across batches. Because the original Stable Diffusion training procedure used a batch size of 3072, we subsequently reduce the image size to 256 × 256 (and thus the corresponding latent dimension to 32 × 32), in order to be able to increase the batch size to 1536. This increase in batch size has led to better training stability and a significantly improved convergence rate."
>
> In summary, because the diffusion models are trained to regress randomly sampled Gaussian noises and Objaverse is a diverse dataset, the variance in the training loss and the training batch construction could make the gradients very unstable and thus a large batch size would be important for a stable convergence.

---

> > ### Comment · Reviewer_bdx1 · 2023-11-21
> >
> > Thanks for the explanation!

---

> > > ### Author Response · Authors · 2023-11-21
> > >
> > > Thanks for the discussion and we really appreciate your efforts in reading our paper and helping us improve the paper!

---

### Official Review · Reviewer_Q4FP · 2023-10-31

**Soundness:** 3 good
**Presentation:** 4 excellent
**Contribution:** 3 good
**Rating:** 6
**Confidence:** 5

**Summary:**

The authors proposed a synchronized multi-view image diffusion model to generate consistent multi-view images. Compared with the previous Zero123, the authors target at generating images at fixed camera views, and the denoising step of each view is conditioned on the noisy images of all views. A feature volume is introduced as an intermediate structure for aggregating information from multi-view noisy images. Experiments show that the generated multi-view images are significantly more consistent than previous methods.

**Strengths:**

- Consistent multi-view image generation with diffusion model has been a useful and heated research topic in the past months. The authors proposed a practical solution for the task.

- Synchronized diffusion conditioned on multi-view information is new to me and is reasonable.

- The visualization of the generated results is good, and experiments show that the multi-view consistency is much better than the Zero123 baseline.

**Weaknesses:**

I am unclear about several components in the proposed pipeline:

- Multi-view image condition: is it possible to use the denoised multi-view images rather than the noisy images from the last step as a condition to the diffusion model?

- 3D CNN: how is the 3D CNN trained? Is it pre-trained or trained with the attention module together in the finetuning?

- 3D CNN: how will the 3D CNN affect the feature aggregation from multi-view and can we remove this part? It reminds me of the cost volume regularization in stereo/MVS networks. If the 3D CNN is used for volume denoising, again, how about directly using the denoised multi-view images from the last step as a condition of the diffusion model?

- View frustum volume: will it be too heavy to pass a view frustum volume to the UNet as the multi-view condition? Also, how is W in the view frustum volume determined?

**Questions:**

See above. Generally, I think the paper is of very good quality. I will raise my rating if my concerns can be addressed in the rebuttal.

---

> ### Author Response · Authors · 2023-11-18
>
> Thank you for your helpful comments! Our replies to your questions and revisions to the submission are stated below.
>
> **Q1**: Is it possible to use the denoised multi-view images rather than the noisy images from the last step as a condition to the diffusion model?
>
> **A1**: SyncDreamer generates multiview images simultaneously in one reverse process, which means all images are denoised at the same time. When denoising one of these images, the other images are also in the generation process and are still noisy so we can only condition on their noisy states. If we adopt an alternative way to generate images sequentially, we can condition on the previously denoised image. However, the sequential generation process can easily suffer from error accumulation and lack of global consistency.
>
> **Q2**: How is the 3D CNN trained?
>
> **A2**: The 3D CNN is not pretrained but is trained together in finetuning the diffusion models to generate multiview images.
>
> **Q3**: 1) How will the 3D CNN affect the feature aggregation from multi-view and can we remove this part? It reminds me of the cost volume regularization in stereo/MVS networks. 2) If the 3D CNN is used for volume denoising, again, how about directly using the denoised multi-view images from the last step as a condition of the diffusion model?
>
> **A3**: 1) Yes, the 3D CNN refines the unprojected features similar to MVSNet. 2) We denoise the multiview images but not the 3D volume. The multiview image denoising process is conditioned on the features provided by the 3D volume so we cannot use denoised images to construct the 3D volume. If we use sequential image generation, we could condition on denoised images but it easily suffers from error accumulations.
>
> **Q4**: Will it be too heavy to pass a view frustum volume to the UNet as the multi-view condition? Also, how is W in the view frustum volume determined?
>
> **A4**: Thanks for your suggestion and we added the description of the volume size in the implementation details (Sec 1.1) of the supplementary material. The spatial volume has the size of $32^3$ and the view-frustum volume has the size of $32\times 32 \times 48$. We sample 48 depth planes for the view-frustum volume because the view may look into the volume from the diagonal direction. The choice of these sizes is based on the consideration that the latent feature map size of a $256\times 256$ image in the Stable Diffusion is exactly $32\times 32$. These volumes indeed bring additional memory and computation consumption, but the overall cost is still acceptable based on our experiments.

---

### Author Response · Authors · 2023-11-18

We thank all reviewers for their efforts in reading our manuscripts and providing insightful feedback. We have revised the main paper and the supplementary material according to your comments, which all are highlighted in magenta.

---

### Meta-Review · Area_Chair_6cVK · 2023-12-04

**Metareview:**

The submission presents a diffusion model called SyncDreamer that generates multiview-consistent images from a single-view image. Compared with the previous Zero123, the authors target at generating images at fixed camera views, and the denoising step of each view is conditioned on the noisy images of all views. A feature volume is introduced as an intermediate structure for aggregating information from multi-view noisy images. Experiments show that the generated multi-view images are significantly more consistent than previous methods.

Pros
* A practical solution to multivew-consistent image genreation
* Great visualization of results

Cons
* The technical design may be suboptimal
* The model has to freeze the pre-trained Zero123 to preserve the original 3d priors, which may indicate flawed designs

**Justification For Why Not Higher Score:**

The main contribution comes from combining different techniques and making the system work. There are some concerns about the technical design about the proposed method.

**Justification For Why Not Lower Score:**

All five reviewers are positive about the submission, with some strongly championing the paper.

---

### Decision · Program_Chairs · 2024-01-16

Accept (spotlight)